# Mitochondrial dysfunction induces ALK5-SMAD2-mediated hypovascularization and arteriovenous malformations in mouse retinas

Haifeng Zhang[1,7], Busu Li [1,7], Qunhua Huang[1,7], Francesc López-Giráldez [2], Yoshiaki Tanaka [3,4], Qun Lin [1], Sameet Mehta [2], Guilin Wang[2], Morven Graham [5], Xinran Liu[5], In-Hyun Park [2], Anne Eichmann[6], Wang Min [1] ✉ & Jenny Huanjiao Zhou [1] ✉

Although mitochondrial activity is critical for angiogenesis, its mechanism is not entirely clear. Here we show that mice with endothelial deficiency of any one of the three nuclear genes encoding for mitochondrial proteins, transcriptional factor (TFAM), respiratory complex IV component (COX10), or redox protein thioredoxin 2 (TRX2), exhibit retarded retinal vessel growth and arteriovenous malformations (AVM). Single-cell RNA-seq analyses indicate that retinal ECs from the three mutant mice have increased TGFβ signaling and altered gene expressions associated with vascular maturation and extracellular matrix, correlating with vascular malformation and increased basement membrane thickening in microvesels of mutant retinas. Mechanistic studies suggest that mitochondrial dysfunction from *Tfam*, *Cox10*, or *Trx2* depletion induces a mitochondrial localization and MAPKs-mediated phosphorylation of SMAD2, leading to enhanced ALK5-SMAD2 signaling. Importantly, pharmacological blockade of ALK5 signaling or genetic deficiency of SMAD2 prevented retinal vessel growth retardation and AVM in all three mutant mice. Our studies uncover a novel mechanism whereby mitochondrial dysfunction via the ALK5-SMAD2 signaling induces retinal vascular malformations, and have therapeutic values for the alleviation of angiogenesis-associated human retinal diseases.

Angiogenesis is the process of blood vessel formation in which proliferating endothelial cells (ECs) sprout from preexisting vessels to extend a vascular network[1,2]. Sprouting angiogenesis is regulated by EC proliferation, migration and metabolic activities that occur in response to environmental cues, such as nutrition deprivation, hypoxia, and inflammation. Angiogenesis is tightly controlled[1,3,4]. Impaired or excessive angiogenesis is associated with the pathogenesis of human retina diseases such as familial exudative vitreoretinopathy (FEVR),

retinal arteriovenous malformations[5–10], diabetic retinopathy (DR) and age-related macular degeneration (AMD)[11,12].

The murine retina provides a good model for the analysis of sprouting angiogenesis. The retinal vasculature invades the eye through the optic nerve head and spreads toward the periphery along the ganglion cell layer during the first postnatal week (P1–P7) to form the superficial vascular plexus. Beginning at P8, the primary plexi sprout perpendicularly or migrate toward the deeper retina, to form

the intermediate and deep vascular plexi[13]. Several angiogenic signaling cascades, including VEGF-VEGFRs and Dll4-Notch, are the best characterized to regulate sprouting tip cell migration and stalk cell proliferation[1,13,14]. It is now well established that angiogenic ECs primarily use glycolysis pathway[15]. While glycolysis is a major ATP source in ECs, mitochondrial respiration-dependent ATP production is thought to play a marginal role in EC metabolism. However, recent data suggest that mitochondrial activity is critical for the biosynthetic and bioenergetic demands that are required for angiogenesis. Specifically, mitochondrial respiration, by providing the building blocks and metabolites necessary for EC proliferation rather than producing ATP, regulates angiogenesis at various steps. Coutelle et al. showed that mice harboring defects in the proofreading function of the catalytic subunit of mitochondrial DNA (mtDNA) polymerase (POLGA) had an altered angiogenesis[16]. The genetic deletion of fatty acid metabolizing enzyme mitochondrial carnitine palmitoyltransferase 1 (*Cpt*1) attenuated the formation and functioning of tip cells[17]. Similarly, the loss of a subunit in complex III (QPC) caused a reduction in EC respiration that impaired cell proliferation and angiogenesis in mice[18]. Recently, Schiffmann et al. showed that mitochondrial respiration is important for EC growth and angiogenesis in embryonic development, wound healing and tumor growth in mice with EC-specific *Cox*10 deficiency[19]. These studies support the pivotal role of mitochondrial activity in angiogenesis.

However, how exactly various mitochondrial activities regulate angiogenic sprouting and proliferation is unclear. In the present study, we explore if distinct mitochondrial activities regulate common EC signaling pathways during angiogenesis. To this end, we investigated retinal angiogenesis in mice with an inducible EC-specific deletion of three separate nuclear genes that encode for mitochondrial proteins responsible for distinct mitochondrial functions, including TFAM, which is required for mitochondrial DNA replication[20], COX10, which is essential for respiration[21], and TRX2, which is critical for the mitochondrial redox state[22]. Our study reveals that distinct mitochondrial activities regulate common signaling pathways that impact retinal vascular development and remodeling.

## Results

### Mitochondrial activity is critical for EC sprouting in 3D sprouting models

To determine mitochondrial function in angiogenesis, we silenced *Tfam*, *Cox10*, and *Trx2* in ECs. Deletion of these three genes had distinct effects on EC mitochondrial DNA copy, ROS generation and respiration (Fig. 1a–i). Specifically, (1) mitochondrial DNA copy was reduced in *Tfam*-KD, but not in *Cox10*- or *Trx2*-KD ECs (Fig. 1b); (2) *Trx2*-KD resulted in a 5.0-fold increase in mitochondrial ROS (mtROS) generation as measured by MitoSOX (Fig. 1c); (3) mitochondrial ATP contents were reduced in all mutant ECs as measured by a bioluminescent assay (Fig. 1d); (4) the Seahorse analyses showed that all mutant ECs decreased the basal oxygen consumption rate, the ATP-linked oxygen consumption, and the reserve capacity (Fig. 1e–i). We then examined the effects of silencing *Tfam*, *Cox10*, and *Trx2* on EC proliferation and migration, two critical processes involved in sprouting angiogenesis. Silencing of *Tfam*, *Cox10*, or *Trx2* attenuated EC proliferation as measured by 5-Ethynyl-2′-deoxyuridine (EdU) incorporation assays (Fig. 2a–b), and EC migration as measured by transwell assays (Fig. 2c–e). To correlate EC proliferation and migration to EC sprouting, siRNA-transfected ECs (with stably expressed EGFP) were applied to a 3D sprouting assay. Silencing of *Tfam*, *Cox10*, or *Trx2* significantly attenuated sprouting as indicated by the reduced number of sprouts and mean sprout length. Consistent with ROS generation in *Trx2*-deficient ECs, ROS scavenger mito-TEMPO rescued *Trx2* siRNA, but not *Tfam* or *Cox10* siRNA impaired-sprouting (Fig. 2f–h). Live/dead cell staining indicated that silencing of *Tfam*, *Cox10*, or *Trx2* did not induce significant cell death

compared to the WT control during the sprouting assay (Supplementary Fig. 1a–b).

Inhibitors targeting various mitochondrial activities, including rotenone on complex I, antimycin on complex III, oligomycin on complex IV also impaired EC sprouting in sprouting assays (Fig. 2i). Quantification analyses indicated that the inhibitors attenuated both average sprout number per spheroid and mean sprout length (Fig. 2j, k).

Taken together, these data suggest that mitochondrial activities are critical for EC sprouting, and are correlated with EC respiration, EC proliferation, and migration.

### Mitochondrial activity is critical for vessel sprouting and maturation in mice

To study the role of mitochondrial proteins in vivo, we generated an inducible EC-specific inactivation for TFAM (*Tfam*^ECKO), COX10 (*Cox10*^ECKO), and TRX2 (*Trx2*^ECKO), respectively. Mitochondrial proteins were expressed in retinal ECs of the control floxed mice with enhanced expression in the angiogenic front of retinas, including tip cells. However, the mitochondrial proteins TFAM, COX10, and TRX2 were depleted in vessels of the respective EC-deficient retinas (Supplementary Fig. 2a). Interestingly, we observed that the mitochondrial proteins were highly abundant in the diving cells toward the deep layer, with less expression in the central vasculature. Mitochondrial proteins TFAM, COX10, and TRX2 were not detected in vessels of respective EC-deficient retinas. Notably, positive staining was visible in neuron cells underneath, suggesting EC-specific deletion in the mutant retinas (Supplementary Fig. 2b). These data suggest that mitochondria might be important for vascular EC sprouting toward the peripheral plexi and diving toward deep layers.

The control floxed mice showed similar vascular progression as visualized by isolectin B4 (IB4) whole-mount staining (Supplementary Fig. 2c–d). However, deletion of *Tfam*, *Cox10*, or *Trx2* resulted in retarded vessel growth in the radial extension of the vascular plexus from the optic nerve head to the periphery (Fig. 3a, b). Abnormal tip cell morphology was evident in mutant retinas, with mutant sprouts displaying short shafts as well as fewer and shorter distal filopodia as visualized by IB4 staining (Fig. 3a–c). The tips cells were further visualized by a tip cell-specific marker ESM1 with CD31 and a reduced numbers of tip cells were obtained from the mutant retinas (Fig. 3d, e). Moreover, a significant reduction (40–60%) in the vascular density (vascular area/total area) in the mutant retinas compared with WT (Fig. 3d, f).

We then examined EC proliferation in *Tfam*^ECKO, *Cox10*^ECKO, and *Trx2*^ECKO retinas at P6 when the retinal vessels were active in sprouting. Proliferative ECs were detected by EdU incorporation, which visualized DNA replication, followed by co-staining with an EC-specific nuclear marker ERG. The EdU⁺ERG⁺ ECs were detected abundantly in the WT, primarily in the front region of the retina. However, the number of EdU⁺ERG⁺ ECs were significantly reduced in the mutant retinas compared to WT when measured for EC density (ERG⁺/mm² retinal area) and EC proliferation index (EdU⁺ERG⁺/ERG⁺ ECs) (Fig. 3g–i). These data indicate that deletion of *Tfam*, *Cox10*, or *Trx2* delays retinal angiogenesis, correlating with reduced EC proliferation which is critical for retinal angiogenesis[23].

Trx2 functions as an endogenous antioxidant that modulates the cellular redox status to regulate cell proliferation and death, and mtROS contributes significantly to the phenotypes observed in various tissues of *Trx2*-deficient mice[22,24]. In contrast, loss of *Tfam* or *Cox10* may attenuate basal ROS due to reduced mitochondrial respiration[20,25]. To visualize mtROS in the vasculature of fresh tissues, vessels were perfused with a high molecular weight FITC-dextran (2000 kDa) by retro-orbital injection together with a mtROS indicator MitoSOX[22,24]. Consistent with our in vitro findings, mtROS were detected abundantly in *Trx2*^ECKO retinas

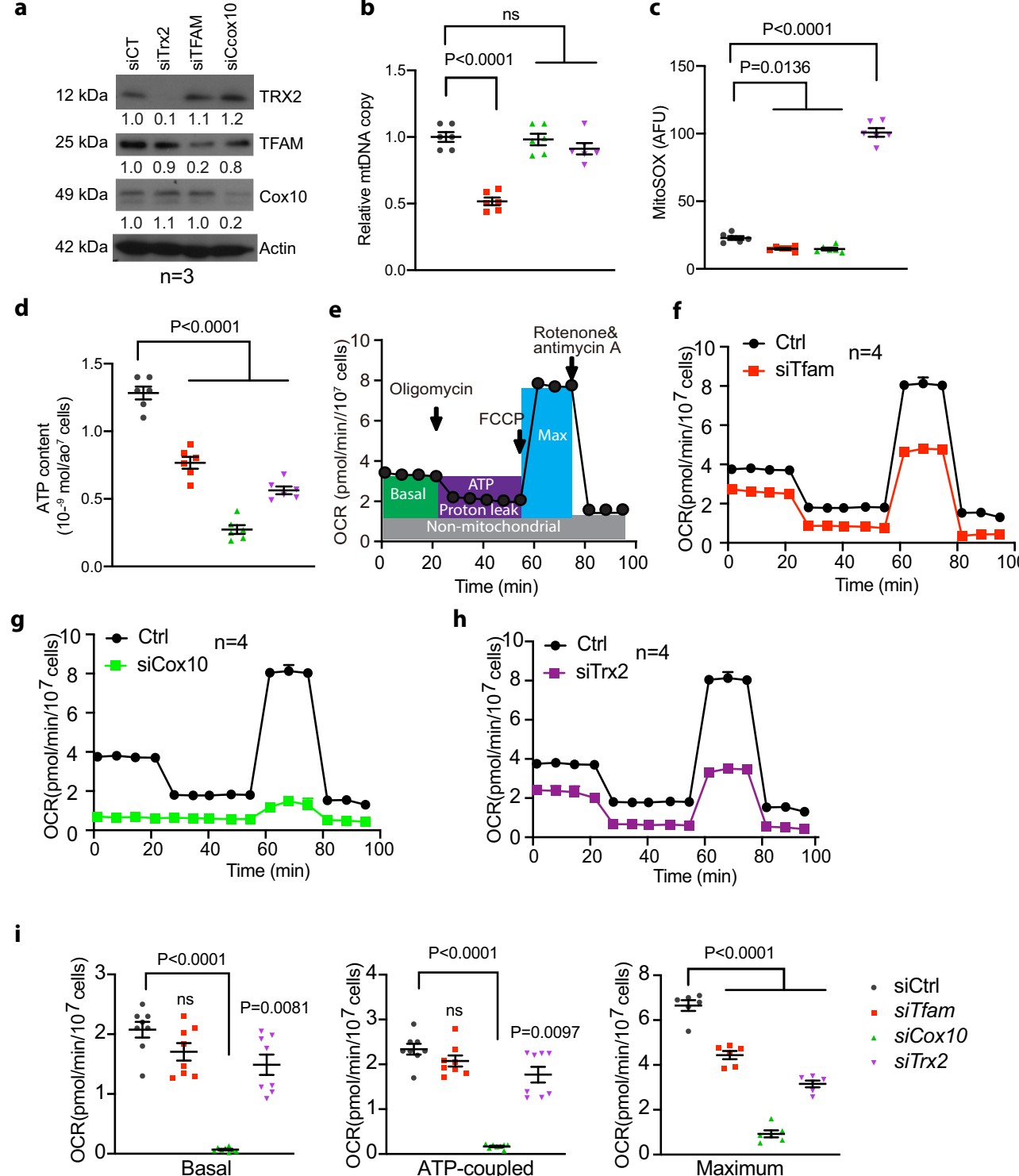

**Fig. 1 | Distinct effects of *Tfam*, *Cox10* and *Trx2* silencing on mitochondrial function.** HUVECs were transfected with control, *Tfam*, *Cox10* or *Trx2* siRNAs. 48 h after transfection, cells were subjected to various assays. **a** Cell lysates were analyzed by Western blotting for TFAM, COX10 and TRX2 depletion with respective antibodies. Protein levels were quantified and presented as fold changes by taking control siRNA as 1.0. $n = 3$ (three independent experiments). **b** Mitochondrial DNA contents were quantitated by qPCR and presented as fold changes compared to control siRNA, $n = 6$ (duplicates from three independent experiments). **c** Mitochondrial ROS (mtROS) were assessed by a mitochondrial-specific ROS probe mitoSOX and fluorescence intensity was presented as arbitrary fluorescence unit (AFU). $n = 6$. **d** ATP production was measured and presented as mmol/cell, $n = 6$ (duplicates from three independent experiments). **e–i** The oxygen

consumption rate (OCR) measured by Seahorse. **e** A diagram for typical mitochondrial stress test profiles. The addition of the coupled respiration inhibitor oligomycin (1 μM) was used to assess ATP production and proton leak. Maximal respiration was measured by adding 1 μM carbonyl cyanide-4-(trifluoromethoxy) phenylhydrazone (FCCP), and spare respiratory capacity and non-mitochondrial respiration was measured by adding 0.5 μM rotenone and 0.5 μM antimycin A. **f–h** OCR in siRNA-transfected HUVECs. **i** The basal, the ATP-coupled and the maximum oxygen consumption rate were calculated. $n = 6$ (Duplicates from three independent experiments). Data are means ± SEM. *P* values are indicated, using one-way ANOVA followed by Tukey's multiple comparisons test. ns: non-significance ($P > 0.05$). Source data are provided as a Source Data file.

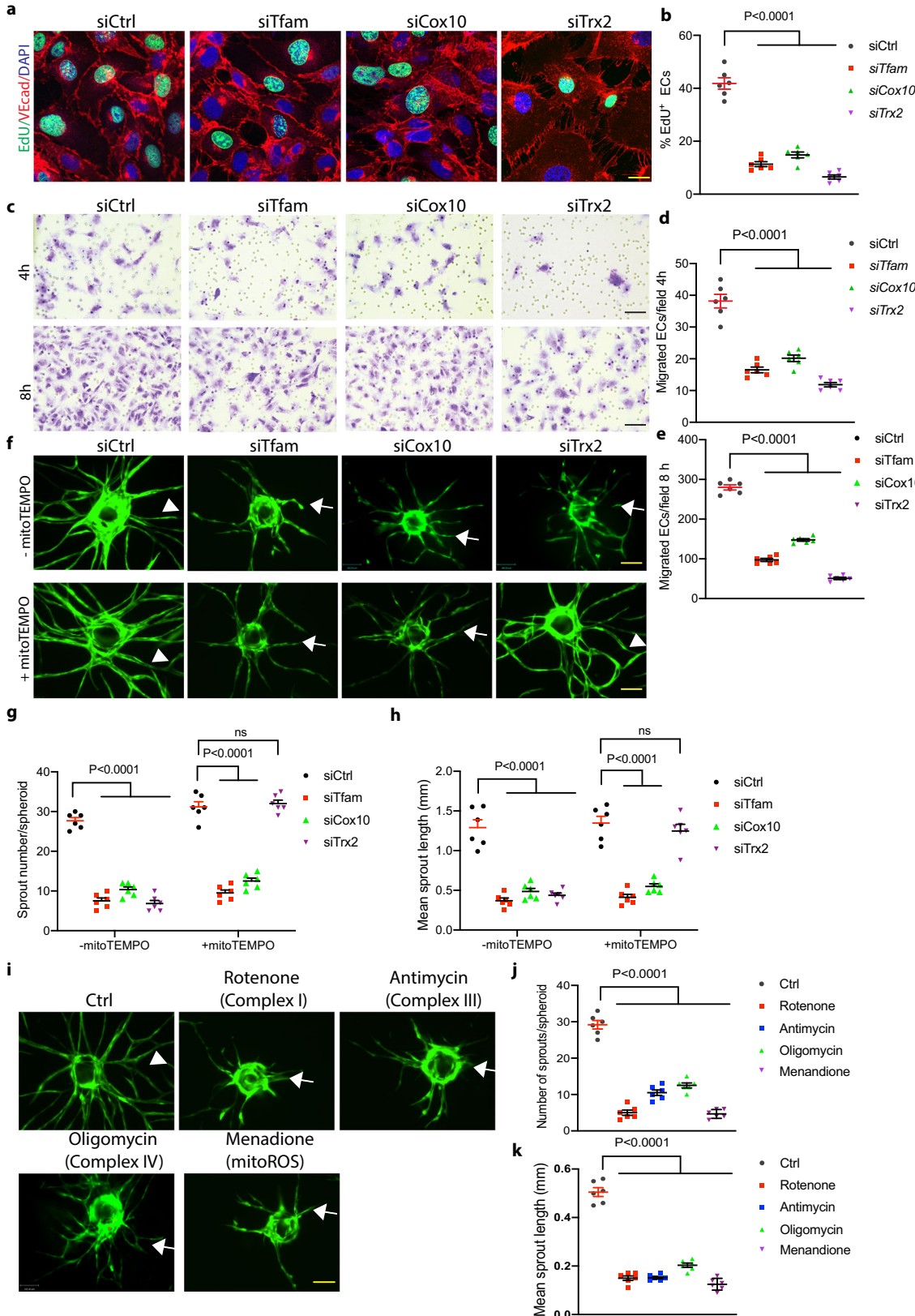

inside and outside the vessels, consistent with the notion that ROS can leak and induce oxidative stress in neighboring cells or tissue[26,27]. However, mtROS were barely detected in WT, *Tfam*[ECKO], or *Cox10*[ECKO] retinas (Supplementary Fig. 3a–c). Therefore, mtROS do not represent a common EC signaling mediating the angiogenic defects caused by mitochondrial dysfunction.

## *Tfam*[ECKO], *Cox10*[ECKO], and *Trx2*[ECKO] retinas exhibit microaneurysm and arterialization at advanced ages

The retinal ECs begin to migrate into the deep layers around P8 to form three layers of vascular network around P12-21, i.e. the surface ganglier cell layer (GCL), the intermediate inner plexus layer (IPL), and the deep inner nuclear layer(INL)/outer plexus layer (OPL)[13]. The mutant retinas

**Fig. 2 | Mitochondrial activity is critical for EC sprouting in 3D sprouting models.** HUVECs were transfected with control, *Tfam*, *Cox10* or *Trx2* siRNAs. 48 h after transfection, cells were subjected to various assays. **a**, **b** EdU incorporation assays for EC proliferation. ECs were stained with anti-VE-cad with counterstaining by DAPI while EdU was detected by a Click-iT assay. % EdU⁺ ECs were quantified (**b**). 10 random fields were counted for each group. Data are duplicates from and three independent experiments. **c**–**e** Cells were plated in fibronectin-coated Transwell filter and incubated for 4 h and 8 h. Migrated cells were visualized by crystal violet staining and quantified. 10 random fields were counted for each group. Data are duplicates from and three independent experiments. **f**–**h** GFP-expressing HUVECs were transfected with control, *Tfam*, *Cox10* or *Trx2* siRNAs. 48 h after transfection, cells were applied to spheroid sprouting assays in the absence or presence of mito-TEMPO (10 µM). Quantification of the sprout number per spheroid (**g**) and mean sprout lengths (**h**) on day 7. 10 spheroids per sample were counted. Data are duplicates from and three independent experiments. **l**, **k** GFP-expressing HUVEC spheroids were cultured in the presence of rotenone (2.5 µM), antimycin (2.5 µM), oligomycin (5 µM) or menadione (1 µM). Quantification of the sprout number per spheroid (**j**) and mean sprout lengths (**k**) on day 7. 10 spheroids per sample were counted. Data are duplicates from and three independent experiments. Data are duplicates from and three independent experiments. Data are means ± SEM. *P* values are indicated, using one-way ANOVA followed by Tukey's multiple comparisons test (**b**, **d**, **e**, **j**, **k**) or two-way ANOVA followed by Sidak's multiple comparisons test (**g**, **h**). Scale bar: 10 µm (**a**); 50 µm (**c**, **f**, **i**). Source data are provided as a Source Data file.

contained avascular areas in the primary plexus, less vascular density and vascular malformation with an arteriovenous shunt at peripheral regions at P15. Moreover, co-staining of IB4 with collagen IV showed a complete colocalization of IB4/Col IV without empty Col IV sleeves (Supplementary Fig. 4a–c). These data suggest that the mutant retinal vessels exhibited retarded sprouting without sprout retraction or vessel collapse after growth.

We then examined the intraretinal vascular phenotype in the P15 mutant retinas by confocal imaging analyses of the different layers. Retinal vessels in WT mice had multiple layers of vessels. However, the mutant retinas contained only the surface vasculature and exhibited a dramatic reduction in branch points with arterial-venous shunts compared to the normal vascular patterning in WT (Fig. 4a, c). This was further verified by presenting the surface GCL, the intermediate IPL and the deep INL/OPL layers individually with each layer pseudo-colored by red, blue and green, respectively. Notably, the stalled diving vessels formed microaneurysm primarily at the surface (Fig. 4b–c). Consistently, there was no detectable intraretinal vessels in the mutant retinas as measured by immunostaining of retinal sections, and microaneurysms were located at the surface with protrusion towards the adjacent immediate layer (Fig. 4d–f). This suggests that retinal vascular ECs could not migrate into deeper layers in the mutant retinas, suggesting that mitochondrial activities are required for intraretinal migration. Notably, the retinas were thinner in the mutant mice compared to the WT mice, consistent with that vascular diving into the inner retina occurs at the same time as neuronal activity begins[28].

The observation of arterial-venous shunts in the mutant retinas prompted us to investigate if arterial-venous identify had been altered. To this end, WT and mutant retinas were subjected to whole-mount staining with venule and arterial markers. Results showed that SMA positive vessels were significantly increased in the mutant retinal veins, which was indicative of arterialization in these vessels (Supplementary Fig. 5a, b; Fig. 4g–i). The increased arterialization in the mutant mice was confirmed with an intrinsic arterial marker Sox17 (Supplementary Fig. 5c, d). Taken together, deletion of *Tfam*, *Cox10*, or *Trx2* induced vessel growth retardation at early stages and vascular malformation at advanced ages.

## High resolution images visualize mitochondrial abnormality in retinal microvessels of mutant mice

We further performed high resolution imaging analyses to examine defects in the mutant retinal microvessels. An increased diameter of the capillary vessels in the mutant retinas was detected (Fig. 5a, b). Notably, the SMA⁺ microvessels were increased in the mutant retinas (Fig. 5c). Endothelial gain of SMA expression, a process of endothelial-mesenchymal transition (EndMT), has been implicated in many pathological diseases[29]. Super resolution stimulated emission depletion (STED) microscopy analyses showed that the SMA⁺ cells in the microvessels of *Tfam*^ECKO^ were mostly NG2⁺ pericytes with few CD31⁺ ECs (Fig. 5d–f), suggesting a minimum of EndMT occurred in the mutant retinas.

Increased extracellular matrix with thickening of basement membrane surrounding capillaries is detected in human retinal microaneurysm[30]. Indeed, electron microscopy (EM) image analyses revealed that capillaries in the mutant retinas exhibited significantly increased basement membrane thickness compared to WT (Fig. 5g, h, j), Notably, pericytes (PCs) in WT were tightly associated with EC with elongated foot processes[31]. However, mutant microvessels showed more rounded PC with shortened foot processes (Fig. 5g, h).

EM image analyses revealed altered mitochondrial numbers and structures. Total number of mitochondria were increased in the mutant retinal ECs (Fig. 5h, i, k). However, mitochondria in retinal ECs of the mutant mice had disrupted cristae with reduced ratio of cristae surface area/outer membrane surface area, indicative of mitochondrial dysfunction in these mutant ECs (Fig. 5h, i, l). There were no obvious apoptotic ECs characterized by blebbing nuclei or mitophagy characterized by typical autophagic vacuoles engulfing damaged mitochondria[22,24,32]. Consistently, mitochondrial DNA copy was reduced in isolated retinal ECs from *Tfam*^ECKO^ mice, but not in *Cox10*- or *Trx2*-deficient retinal ECs. ATP contents were reduced in all mutant ECs (Fig. 5m, n). Taken together, our data suggest that the angiogenic retardation in the mutant mice was due to mitochondrial dysfunction rather than as a result of massive cellular collapse.

## Single-cell RNA-seq analyses reveal altered TGFβ signaling associated with increased extracellular matrix and reduced vessel maturation

The common AVM phenotype in the mutant mice prompted us to further examine its underlying mechanisms. To this end, we performed single-cell (sc)RNA-seq transcriptome analyses of P12 retinal ECs. The WT had more CD31⁺CD45⁻ ECs collected compared to the mutant retinas but similar median genes per cell were obtained in all four groups (Supplementary Fig. 6a, b). A total of 10 clusters were identified using unsupervised graph-based clustering (Fig. 6a). The positive differentially expressed genes for each cluster compared to all other cells were analyzed (Fig. 6b dotplot for specific marker genes; Supplementary Table 1 for top 50 genes). Two smaller segregated clusters were identified as pericytes using *Pdgfrb* and *Cspg*4 genes (PC; cluster 8) and photoreceptor cells using *Gngt*1 and *Sag* genes (Cluster 9). Both clusters were excluded from further analyses. The seven major EC clusters were subjected to gene ontology (GO) analyses (Fig. 6c for the representative GO pathways; Supplementary Table 2 for top 50 GO list). Cluster 0 cells were enriched for EC development and vasculogenesis-related genes (named as EC development). Cluster 1 cells were enriched for transcriptional factors, including *Smad*-regulatory genes (named as Transcriptional factors). Cluster 2 cells were enriched for ribosomal subunits and ribosome biogenesis-related genes. Besides the ribosomal genes, a group of respiration-related genes were specifically altered. We therefore named this cluster as Respirasome. Cluster 3 cells were enriched for transporter genes (named as Transporter). Cluster 4 cells were enriched for extracellular matrix (ECM)-related genes (named as ECM). Cluster 5 cells were enriched for angiogenic and tip cell-associated genes. Cluster 6 were

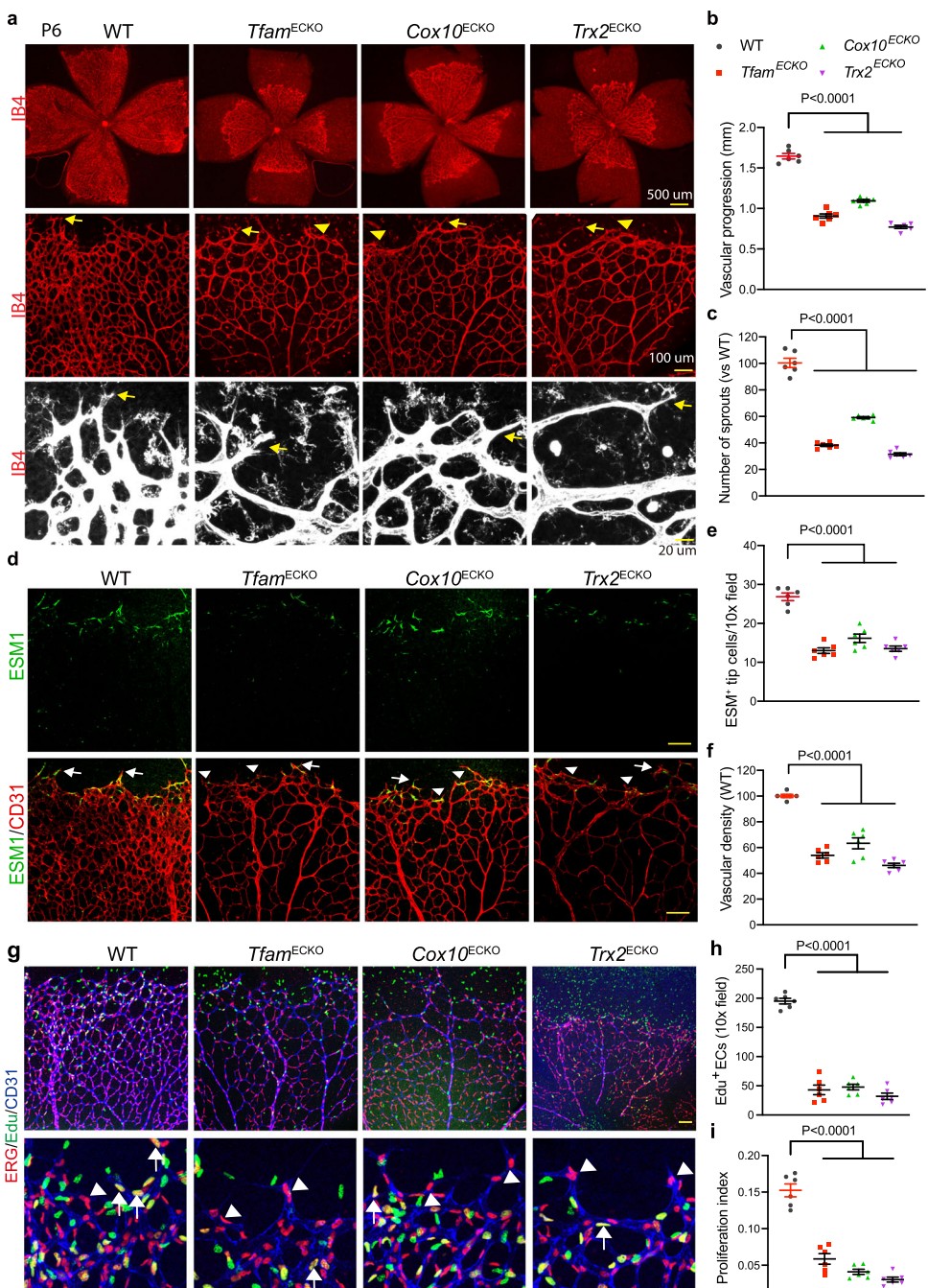

**Fig. 3 | Mitochondrial activity is critical for retinal sprouting in mice. a–c** P6 retinas from WT, *Tfam*^ECKO, *Cox10*^ECKO and *Trx2*^ECKO were subjected to whole-mount staining for IB4. **a** Different power images are shown. Arrows and arrowheads indicate normal and defective sprouts, respectively. **b–c** Vascular progression (the distance of radial extension of the vascular plexus from the optic center) and number of sprouts was quantified. *n* = 6 mice per group. **d–f** Visualization of tip cells. P6 retinas were subjected to whole-mount staining for IB4 with ESM1. Arrows and arrowheads indicate normal and defective tip cells, respectively. ESM1⁺ Tip cell number (**e**) and relative vascular density (**f**) were quantified. *n* = 6 mice per group. **g–i** WT, *Tfam*^ECKO, *Cox10*^ECKO and *Trx2*^ECKO were received intraperitoneal injection of EdU (100 µg per gram of body weight). At 4 h, retinas were harvested and subjected to whole mount staining with ERG (red) with CD31 (blue) and EdU Click-iT assays (green). Arrows and arrowheads indicate EdU⁺ and EdU⁻ EC, respectively. Total EdU⁺ERG⁺ cells (EdU⁺ EC) and EC proliferation index (EdU⁺ERG⁺/ERG⁺ ECs) were quantified (**h, i**). *n* = 6 mice per group. Data are means ± SEM. *P* values are indicated, using one-way ANOVA followed by Tukey's multiple comparisons test. Scale bar: Three panels in (**a**) are indicated (500 µm, 100 µm and 20 µm, respectively). 100 µm (**d, g** top panel); 20 µm (**g** bottom panel). Source data are provided as a Source Data file.

enriched for RNA splicing (named as RNA splicing), and Cluster 7 cells were enriched for cell cycle-related genes (named as Mitotic). More importantly, Cluster 3 (Transporter) was primarily composed by WT ECs whereas Cluster 4 (ECM) was largely composed by the mutant ECs without WT ECs (Fig. 6d). In addition, the cell number in angiogenic/tip cell cluster (Cluster 5) were reduced in all three mutant ECs compared

to WT (Fig. 6d). These data were consistent with the reduced vessel maturation and the increased basement membrane thickening observed in the mutant mice.

We then performed unbiased comparisons of differential expressed genes (DEGs) between the different mutant mouse models and WT. Specifically, DEGs in the total EC populations (Cluster 0-7

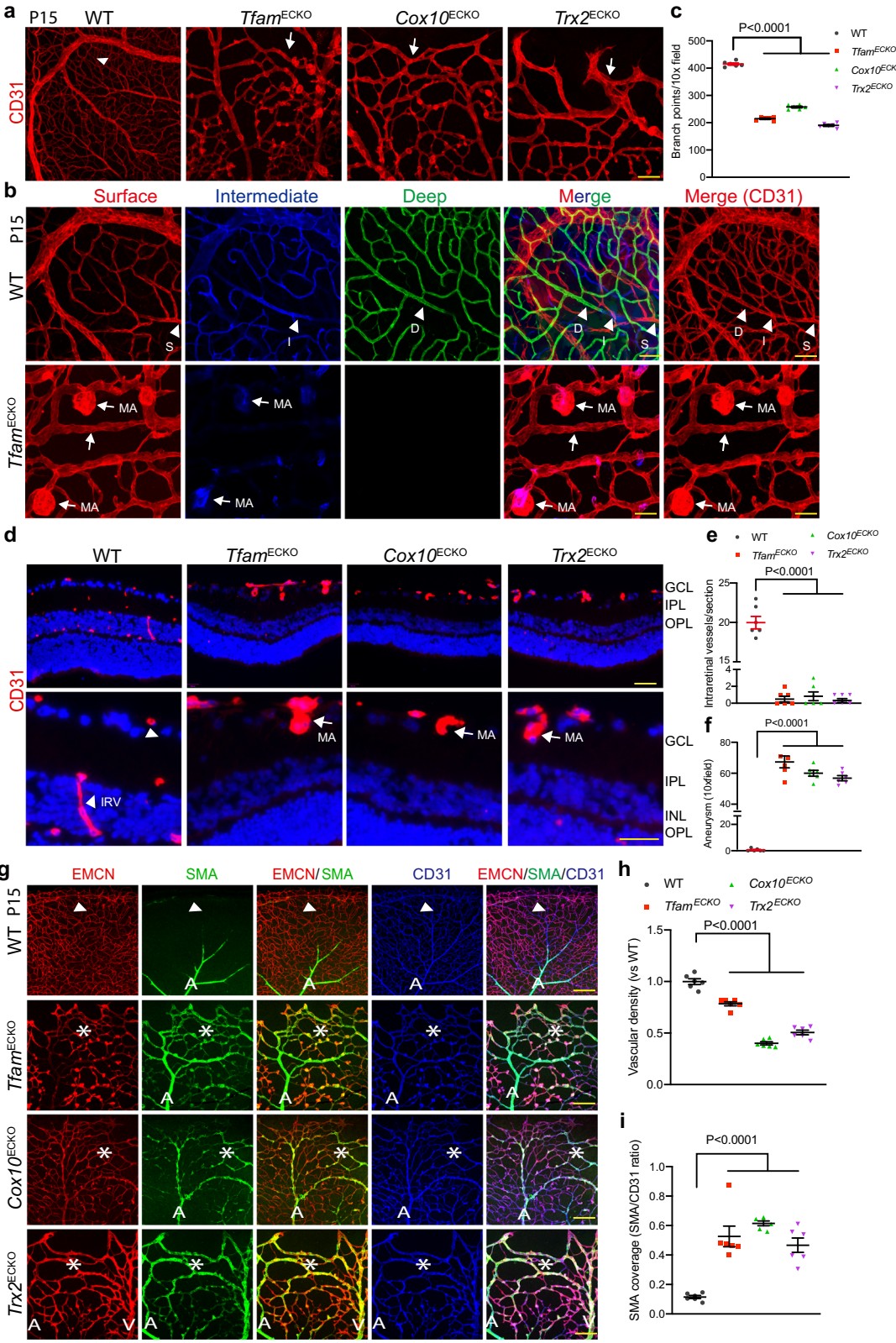

combined together upon removing pericyte and neuron clusters) were compared between each mutant with the WT (see GEO files). We identified 138 common significantly DEGs (Two-sided Wilcoxon Rank Sum test adjusted *p*-values < 0.5, based on Bonferroni correction using the total number of genes, Supplementary Table 3) among all 3 comparisons. Interestingly, these genes were similarly upregulated or

downregulated in the mutant ECs compared to WT (Fig. 6e). Among these 138 genes, transporter genes were drastically downregulated in the mutant ECs (Fig. 6f), consistent with immature vasculature in the mutant retinas. On the other hand, the ECM-associated genes were either highly upregulated or downregulated in the mutant ECs (Fig. 6g). Moreover, we found that the TGFβ regulatory pathway, a

**Fig. 4 | *Tfam*^ECKO^, *Cox10*^ECKO^ and *Trx2*^ECKO^ retinas exhibit microaneurysm and arterialization at advanced ages. a–c** P15 retinas from WT, *Tfam*^ECKO^, *Cox10*^ECKO^ and *Trx2*^ECKO^ treated with TAM at P1 were subjected to whole mount staining for CD31 (**a**). Vasculatures in three retinal layers (surface, intermediate and deep) were imaged by confocal microscopy analyses, and pseudo-colored by red, blue and green, respectively. WT had three layers of vasculature (unique vessels in each layer are indicated by arrows) while the mutant retinas had only the surface vessels with stalled diving vessels (arrowheads) (**b**). Vessel branch points were quantified (**c**). *n* = 6 mice per group. **d–f** Intraretinal vessels from GCL into intermediate IPL and deep INL/OPL layers were examined by CD31 staining of retina cross sections (**d**).

Microaneurysm (**e**) and intraretinal vessels/section (**f**) were quantified. *n* = 6 mice per group. **g–i** P15 retinas were subjected to whole mount co-staining with α-SMA, endomucin (EMCN) and CD31. Arrowheads and asterisks indicate normal α-SMA⁻ microvessels in WT and α-SMA⁺ microvessels in the mutant retinas, respectively. A: artery/arterial; V: vein. Microvessel density (**h**) and α-SMA coverage (α-SMA:CD31) (**i**) were quantified by Image J. *n* = 6 mice per group. Data are means ± SEM. *P* values are indicated, using one-way ANOVA followed by Tukey's multiple comparisons test. Scale bar: 50 μm (**a**, **d**); 25 μm (**b**); 100 μm (**g**). Source data are provided as a Source Data file.

major regulator of ECM-associated genes, was commonly altered in the three mutant ECs (Fig. 6h). In addition, classical sprouting tip cell (S-Tip) genes were upregulated while the diving tip cell genes (D-Tip) genes recently identified[33] were specifically downregulated in the mutant retinas (Fig. 6i). We further determined if the TGFβ signaling was altered in the mutant retinas. Immunostaining indicated that phosphor-SMAD2 was highly upregulated in the retinal ECs in the mutant retinas (Fig. 6j, k). Therefore, our scRNA-seq analyses revealed that altered gene expressions in EC sprouting, vascular maturation, and ECM are tightly associated with the observed phenotypes in the mutant mice. Moreover, the TGFβ–SMAD signaling was commonly upregulated in the three mutant retinal ECs.

### Mitochondrial dysfunction augments TGFβ-ALK5-SMAD2 signaling

Canonical TGFβ signaling[34] is initiated by TGFβ ligand dimer binding to the trans-membrane receptors TβRII and TβRI, forming a receptor tetrameric complex in which TβRI is trans-phosphorylated by TβRII. Activated TβRI phosphorylates downstream R-SMAD (e.g., SMAD2) proteins, resulting in the formation of SMAD2–SMAD4 complexes, which accumulate in the nucleus and act as transcription factors to activate or repress gene transcriptions. The non-canonical PI3K/Akt signaling pathways can be also activated. ALK5 is a major TGFβR1 in ECs that represses EC proliferation and migration, which leads to quiescent state[9,35]. Many proteins involved in TGFβ activation could be transcriptionally induced by TGFβ, including TGFβR1 and SMADs, forming a feedback loop[36–39]. We dissected the mitochondrial effects on the TGFβ signaling in cultured ECs. Silencing of *Tfam*, *Cox10*, or *Trx2* in human ECs (HUVECs) increased phosphor-SMAD2 in the absence of ligand TGFβ. Interestingly, the levels of ALK5 and SMAD2 proteins, but not TGFβR2 or SMAD1, were increased in *Tfam*, *Cox10*, or *Trx2*-silenced ECs (Fig. 7a). Similarly, the mRNA levels of *Alk5* and *Smad2*, but not of *Tgfbr2* or *Smad1* (Fig. 7b). The half-lives of ALK5 and SMAD2 were not significantly altered in the *Tfam*-silencing ECs (Fig.7c), suggesting that they were regulated primarily at a transcriptional level. Immunostaining showed that nuclear SMAD2 was increased in the mutant ECs (Fig. 7d, e). Interestingly, colocalization of phosphor-SMAD2 with mitochondrial marker TOM20 was detected in *Tfam*-silenced ECs by high resolution microscopy (Fig. 7f, g). The enhanced mitochondrial localization of phosphor-SMAD2 upon *Tfam* depletion was further confirmed by immunogold electron microscopy (immuno-EM)[40]. TFAM protein was detected in WT but was absent in *Tfam*-silenced ECs, and phosphor-SMAD2 was evidently translocated from cytosol in WT ECs to mitochondria in *Tfam*-silenced ECs (Fig. 7h, i). Consistent with increased ALK5 expression in the mutant ECs, ALK5-specific inhibitors (LY364947, RepSox and SB535) blunted the TGFβ-induced SMAD2 activation in *Tfam-silenced* ECs. Interestingly, *Tfam* silencing did not augment Akt activation, suggesting that mitochondrial dysfunction specifically activated the canonical TGFβ pathway (Fig. 7j).

SMADs can be directly phosphorylated by MPAKs to enhance their transactivation[36–39]. We detected increased phosphorylations of MAPKs (ERK1/2, p-38 and JNK) in *Tfam*-silenced ECs. Expression of ASK1, an upstream of p38/JNK MAPK that could be activated by mitochondrial dysfunction in *Trx2*-deficient cells[22], was also increased

in *Tfam*-silenced ECs. Furthermore, the activation of ASK1 and MAPKs in *Tfam*-silenced ECs was TGFβ-independent (Fig. 7k). We then examined if ASK1 and MAPKs mediated the upregulation of ALK5 and SMAD2 in *Tfam*-silenced ECs using specific inhibitors. JNK-, p38- and ERK1/2-specific inhibitors, but not ASK1 inhibitor[22], diminished the upregulation of ALK5 and SMAD2 protein levels in *Tfam* silenced ECs (Fig. 7l).

Taken together, these data suggest that mitochondrial dysfunction via activating MAPKs enhances TGFβ-ALK5-SMAD2 signaling.

### ALK5 inhibitor rescued the retarded vessel growth and vascular malformation in the mutant mice

To determine if augmented ALK5 signaling contributed to the vascular defects observed in mutant mice, we first tested the effect of ALK5 inhibitors in a 3D sprouting assay. Silencing of *Tfam* significantly attenuated sprouting as indicated by the reduced number of sprouts and mean sprout length. Inhibition ALK5 signaling by LY364947 partially rescued *Tfam*-siRNA impaired sprouting as indicated by the increased sprout numbers and sprout lengths (Supplementary Fig. 7a, c).

We next examined if inhibition of TGFβR could rescue the defects in vascular sprouting and malformations in the mutant mice. To this end, gene deletion was induced by tamoxifen at P1-3 followed by intraperitoneal injections of ALK5 inhibitor LY364947 daily from P2–P6 or P2–P14 (Fig. 8a for Protocols). LY364947 blocked the augmented TGFβR signaling in *Tfam*^ECKO^ mice as measured by Western blotting for phosphor-SMAD2 (Supplementary Fig. 8a, b) and by co-immunostaining (Supplementary Fig. 8c, d). Deletion of *Tfam* significantly reduced the vascular areas and vascular branches. However, the ALK5 inhibitor LY markedly rescued the retarded retinal vessel growth in the mutant retinas (Fig. 8b, c for *Tfam*^ECKO^; Supplementary Fig. 9a, b for *Cox10*^ECKO^ and *Trx2*^ECKO^).

Confocal imaging analyses showed that the ALK5 inhibitor treatment did not affect the superficial and intraretinal vessel growth in the WT retinas. However, the ALK5 inhibitor rescued the delayed superficial vessel growth and branching as well as intraretinal vessel growth in P15 *Tfam*^ECKO^ mice (Fig. 8d–f; Supplementary Fig. 9c, d for *Cox10*^ECKO^ and *Trx2*^ECKO^). High resolution imaging indicated that ALK5 inhibitor did not affect vascular patterning in WT retinas, which exhibited normal arterial and venule network. However, the ALK5 inhibitor significantly attenuated the arterialization of capillary beds and normalized the vascular pattern in the mutant retinas (Fig. 8g, h). Similarly, the ALK5 inhibitor significantly diminished microaneurysms in the mutant retinas (Fig. 8i).

Taken together, these data suggest that the augmented ALK5 signaling contributed to the retinal growth retardation and vascular malformation in the mutant retinas.

### Genetic deficiency of Smad2 rescued the vascular malformation in Tfam^ECKO^ mice

Finally, we tested if genetic deficiency of *Alk5* or *Smad2* could rescue the phenotype in the mutant mice. Deficiency of *Alk5*, but not of the canonical *Smad* effectors, cause retinal sprouting defects and hemorrhagic vascular malformations[33]. Therefore, we focused

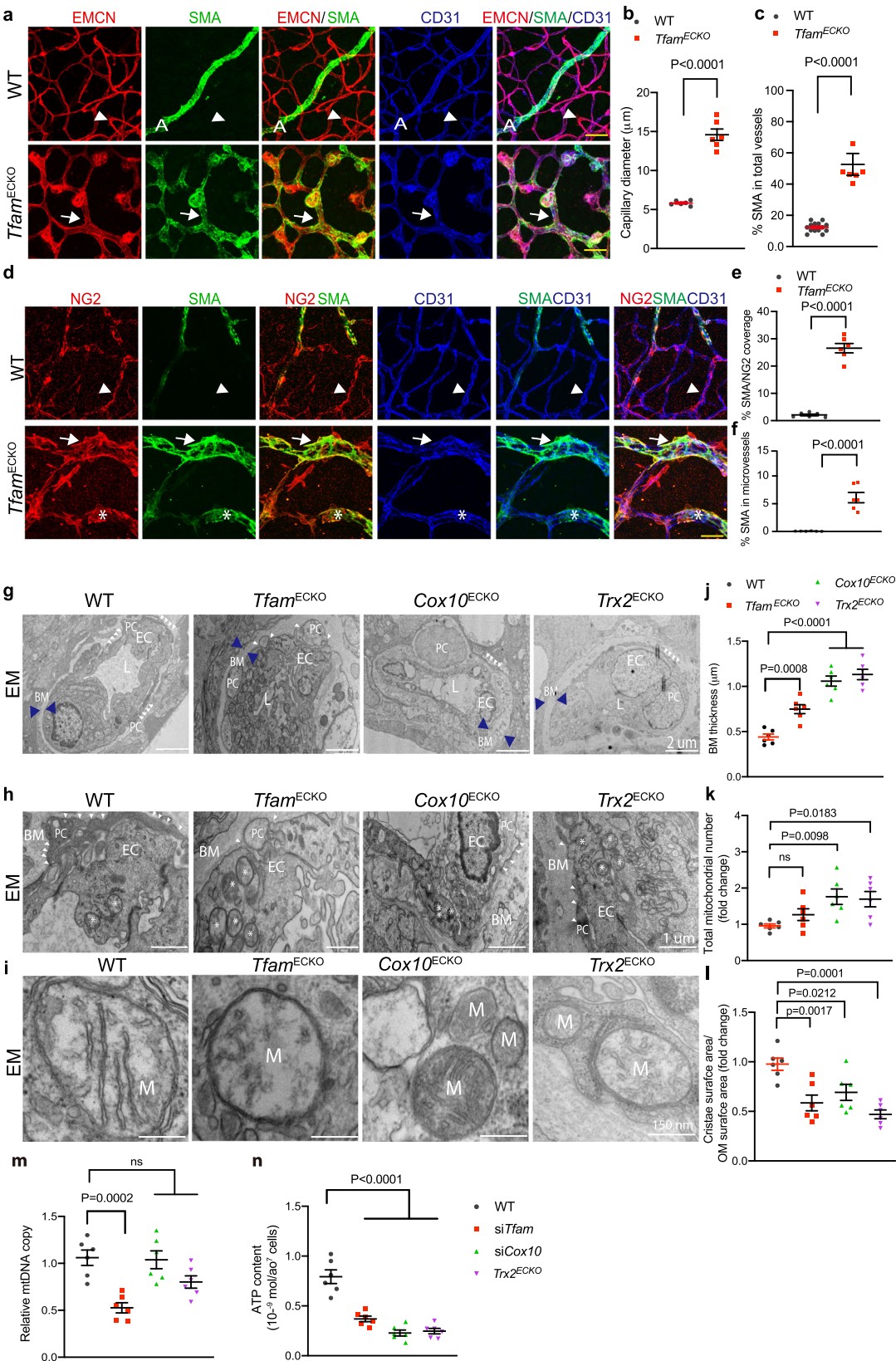

on genetic rescues by the *Smad* deficiency[41]. *Smad2*^ECKO and *Tfam*^ECKO*Smad2*^ECKO (DKO) were obtained by treating Cdh5CreER^T2:*Smad2*^fl/fl and Cdh5CreER^T2:*Tfam*^fl/fl:*Smad2*^fl/fl pups at P1 to P3 with tamoxifen. P15 retinal tissues were examined for sprouting defects and microaneurysms. *Smad2* deletion in retinal ECs was verified by qRT-PCR and Western blotting by comparing ECs vs non-EC populations

from *Smad2*^ECKO retinas (Supplementary Fig. 10a, b). Phosphor-SMAD2 was increased in *Tfam*^ECKO retinas but was diminished in the DKO mice (Supplementary Fig. 10c). No obvious vascular defects were found in *Smad2*^ECKO retinas. However, *Smad2* deletion completely rescued the vascular defects caused by *Tfam* deletion, as measured by the normalized vascular progression, diminished arterialization, and

**Fig. 5 | High resolution images visualize mitochondrial abnormality in retinal microvessels of mutant mice. a–c** P15 retinas were subjected to whole mount co-staining with α-SMA, endomucin (EMCN) and CD31. Arrowheads and arrows indicate normal α-SMA⁻ microvessels in WT and α-SMA⁺ microvessels in the mutant retinas, respectively. A: artery/arterial. Capillary diameter (**b**) and % α-SMA coverage in total vessels (**c**) were quantified by Image J. **d–f** P15 retinas were subjected to whole mount co-staining with NG2, α-SMA and CD31 followed by imaging under STED microscope. Arrowheads and arrows indicate normal α-SMA⁻NG2⁺ microvessels in WT and α-SMA⁺NG2⁺ microvessels in the mutant retinas, respectively. α-SMA⁺CD31⁺ microvessels in the mutant retinas were indicated by asterisks. % α-SMA/NG2 overlap (**e**) and % α-SMA/CD31 overlap (**f**) were quantified by Image J. **g–l** P15 retinas from WT and mutant mice were subjected to EM. **g–i** Representative images of capillaries containing EC surrounded by pericyte (PC), basement membrane (BM) and mitochondria (M). L: capillary lumen. PC processes are indicated by white arrowheads. **j** Mean BM thickness, (**k**) Total mitochondrial number, (**l**) Cristae surface area/outer membrane surface area are quantified. 10 EM section per mouse. $n = 6$ mice per group. Data are means ± SEM. $P$ values are indicated, using one-way ANOVA followed by Tukey's multiple comparisons test. Scale bar: 25 μm (**a**); 10 μm (**d**); 2 μm (**g**); 1 μm (**h**); 150 nm (**i**). Source data are provided as a Source Data file.

normalized microvascular density (Supplementary Fig. 10d–f; Fig. 9a–c). Moreover, *Smad2* deletion completely recovered vascular diving into deep layer, and diminished formation of AVM and microaneurysm in DKO compared to *Tfam*^ECKO mice (Fig. 9d–f). Taken together, these data suggest that increased SMAD2 mediated the mitochondrial dysfunction-induced retinal vascular effects in the mutant mice.

## Discussion

Mitochondrial activity is critical for angiogenesis; however, its mechanism is not fully understood. By examining retinal angiogenesis and transcriptomics in mice with an inducible EC-specific deletion of TFAM, COX10, and TRX2 that are responsible for distinct mitochondrial functions, we reveal that distinct mitochondrial activities can regulate common signaling pathways critical for angiogenesis, including EC sprouting, proliferation, and maturation. More specifically, the silencing of *Tfam, Cox10*, or *Trx2* in ECs causes defects in cell migration, proliferation and angiogenic sprouting using an in vitro 3D sprouting assay; attenuated EC sprouting is correlated with reduced EC respiration and ATP production. Consistently, pharmacological inhibition of complex I, complex III, or complex IV or induction of mtROS impairs EC sprouting. Inducible EC-specific deletion of a single gene *Tfam, Cox10*, or *Trx2* in mice retards retinal vessel growth at early ages (P5–P12) and induces AVM at advanced ages (P15–P30) in these mice. The common phenotypes of *Tfam*^ECKO, *Cox10*^ECKO, and *Trx2*^ECKO mice are correlated with attenuated tip cell sprouting, EC proliferation, but not with ROS production in the mutant mice. The scRNA-seq analyses of retinal ECs reveal common EC populations with altered gene expressions in angiogenic sprouting, vascular maturation, ECM, and TGFβ signaling. In supporting TGFβ signaling in the mutant mice, increased ALK5-SMAD2 activations are observed in the mutant ECs and by immunostaining in the mutant retinal ECs compared to WT retinas. Although a low basal TGFβR signaling is maintained in the tip ECs[42], our study suggests that overactivated TGFβR signaling blocks vascular sprouting and diving. Mechanistically, the MAPKs pathways induced by mitochondrial dysfunction contribute to the observed over-activation of the ALK5-SMAD2 signaling in retinal ECs. Importantly, pharmacological blockade by ALK5 inhibitors or genetic deficiency of SMAD2 attenuates retinal vessel growth retardation and vascular malformation in all the mutant mice. Our study reveals a critical role of the ALK5-SMAD2 signaling in mediating mitochondrial dysfunction-induced vascular defects (Fig. 10a).

The role of glycolysis in tip cell formation and vascular sprouting has been well documented[43,44]. It is proposed that the high amounts of glycolysis-produced ATP in tip cells are needed for actin cytoskeleton remodeling and tip cell competition with neighboring stalk cells[43,44]. Despite the important role of glycolysis in tip cells, the role of mitochondria in tip cell formation has not been directly tested. An important finding in our study is that mitochondrial activity is critical for EC migration and sprouting. This is supported by the following observations: 1) we employed a transwell assay to directly measure EC migration at a single-cell level and the silencing of *Tfam, Cox10*, or *Trx2* attenuates EC migration; 2) in the in vitro 3D sprouting assay, the silencing of *Tfam, Cox10*, or *Trx2* significantly attenuated EC

sprouting; 3) deletion of a single gene of *Tfam, Cox10*, or *Trx2* reduced tip cell formation; 4) deletion of a single gene of *Tfam, Cox10*, or *Trx2* also diminished penetration of inter-retinal vessels. Interestingly, we observed that the mitochondria were highly abundant in tip cells at the angiogenic front and diving cells toward the deep layer, and they were less abundant in the relatively mature central retinal vasculature. The mechanism by which mitochondria control tip cell sprouting is unclear. Mitochondria in ECs have been shown to contribute to mtROS-dependent VEGF production and migration[27]. It is also reported that mtROS are critical for EC polarity and migration by activating the MST1-FOXO1 axis[45]. Therefore, mitochondria may regulate tip cell sprouting via basal ROS production and respiration metabolite-mediated signaling and gene expression[46]. Alternatively, mitochondrial activity may regulate cell migration through cytoskeleton assembly and organization. In support of this model, our scRNA-seq assays identify an EC population were enriched in actin cytoskeleton genes, which were downregulated in the three mutant ECs. These data support the critical function of actin organization in sprouting angiogenesis[47,48]. Additionally, the role of integrin-ECM in cell migration is well documented[49–51], and ECM genes are also altered in the three mutant ECs. More work is needed to define the mechanism by which mitochondria regulate actin organization and ECM remodeling that is associated with EC migration.

A key finding in our study is that the TGFβR signaling is a critical effector that links mitochondrial dysfunction to retarded EC proliferation and sprouting angiogenesis. This conclusion is based on the culmination of multiple pieces of data. (1) scRNA-seq analyses show that genes associated with migration and actin assembling were downregulated in all EC populations. An extensive literature search reveals that these genes are TGFβ responsive (either up or down-regulation). (2) ALK5-dependent SMAD2 phosphorylation is detected in the mutant retinas. (3) The silencing of *Tfam, Cox10*, or *Trx2* gene enhances ALK5-SMAD2 activities which are blocked by ALK5 inhibitors. (4) Furthermore, we uncover that mitochondrial dysfunction via activating MAPKs enhances the ALK5-SMAD2 expression signaling in a ligand-independent manner. We postulate that SMAD2-associated mitochondria can sense the mitochondrial defects and are readily phosphorylated by MAPKs, leading to enhanced SMAD2 nuclear translocation and SMAD2-dependent gene expression. (5) Importantly, ALK5 inhibitors or genetic deletion of *Smad2* in ECs normalizes EC proliferation and vessel growth, and diminished vascular malformation (arterialization and microaneurysm) in the mutant mice. Therefore, the augmented ALK5-SMAD2 signaling in *Tfam, Cox10*, and *Trx2*-deficient ECs drives vascular defects and malformations in the mutant mice (Fig. 10b).

Loss or gain of TGFβ signaling has been implicated in several vascular malformation, such as HHT and cerebral cavernous malformation (CCM). HHT, a genetic bleeding disorder that leads to systemic AVMs in multiple organs, is caused by loss-of-function mutations in the ALK1/ENG/SMAD4 pathway. Mechanism studies suggest that the pathogenesis of HHT strongly relies on overactivation of the non-canonical PI3K/AKT signaling in ECs, which leads to EC proliferation and vascular hyperplasia[7]. Given that the ALK1 signaling pathway can inhibit ALK5-dependent transcriptional activation[9,35], it is unclear whether ALK5-

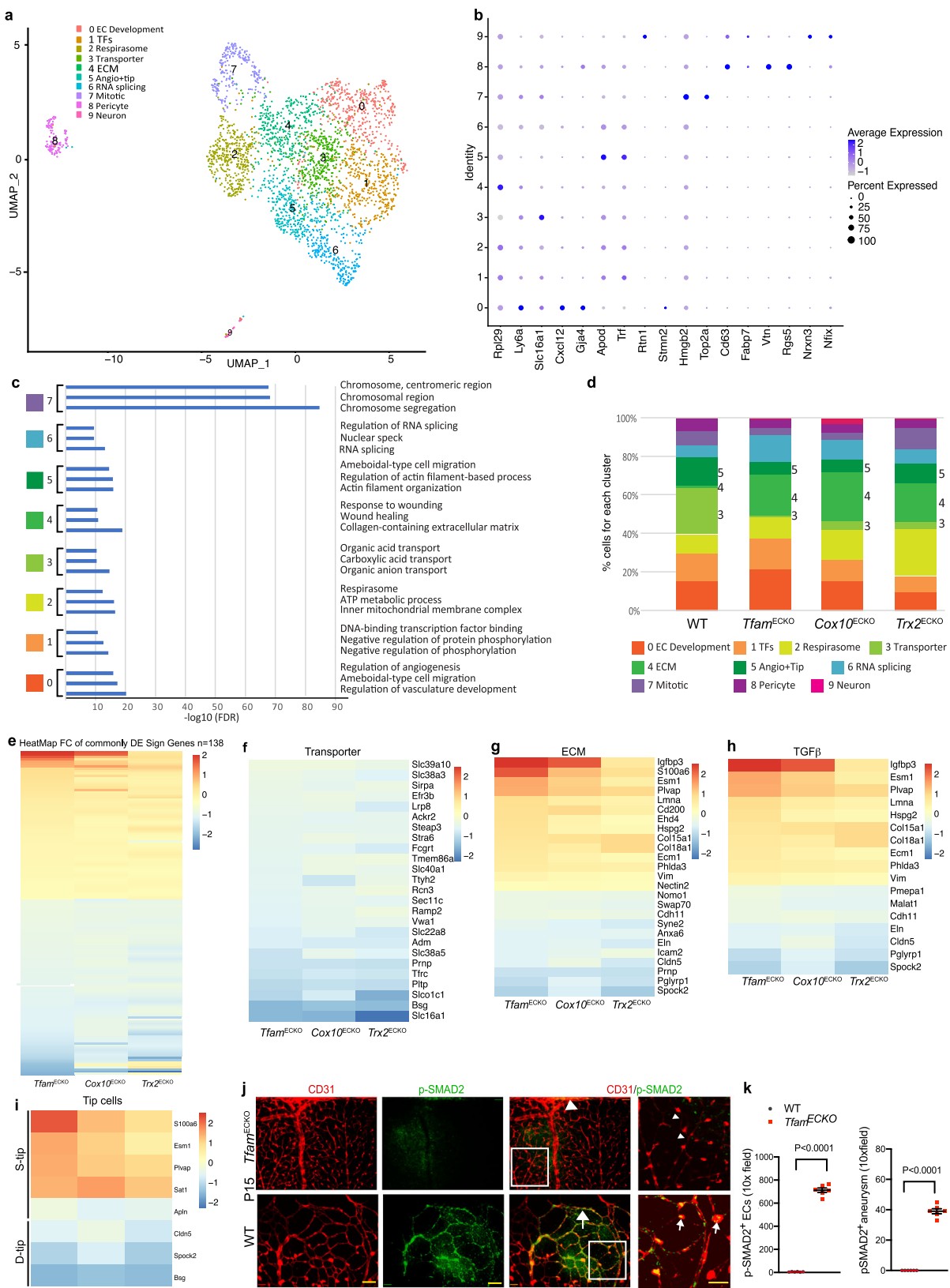

SMAD2 are upregulated in HHT as we observed in our mitochondrial mutant mice. Of note, EC proliferation is enhanced rather than attenuated in HHT models. Furthermore, the AVM in the retinas of mouse HHT models occurs primarily in the central plexus[6,7]. Similarly, augmented EC proliferation contributes to the pathogenesis of CCM, which forms clusters of enlarged capillary endothelial channels ("caverns")

within the vasculature of the central nervous system[52,53]. Interestingly, a gain of TGFβ is involved in CCM progression, which can be blocked by TGFβ inhibitors[9,54]. Therefore, mitochondrial dysfunction-induced AVMs have similarity but distinct features from that of HHT and CCM in the aspects of anatomy of the vasculature, the ALK5-SMAD2 signaling, and the reduction in EC proliferation.

**Fig. 6 | Single-cell RNA-seq analyses reveal gene signatures in the mutant ECs.**
**a** Uniform Manifold Approximation and Projection (UMAP) for single-cell transcriptomes in ECs from WT, Tfam[ECKO], Cox10[ECKO], and Trx2[ECKO] retinas colored accordingly to identified clusters. **b** Dotplots for specific marker genes in each cluster. **c** The representative GO pathways for each cluster. False discovery rate (FDR) was adjusted by p.adjust function in R with "method = "BH"" parameter. **d** Percentage of cells for each cluster in WT and mutant retinas. Clusters 3, 4, and 5 in the mutant retinas were evidently altered compared to WT are indicated. **e** Heatmap of the log2 fold changes of the gene expression of the common differentially expressed genes (Common DEGs) (n = 138, adjusted p-value < 0.05) between each mutant and the WT. **f–i** Heatmaps of the log2 fold changes of specific

groups of genes (Transporters, ECM, and TGFβ signaling) from the 138 DEGs. n = 3 mice. Genes were considered significantly differentially expressed when the two-sided Wilcoxon Rank Sum test adjusted p-value, based on Bonferroni correction using the total number of genes in the dataset, was below 0.05. **j, k** Visualization of p-SMAD2. P15 Tfam[ECKO] retinas were subjected to whole mount co-staining with p-SMAD2 and CD31. **j** Low power images of vascular plexi at top and high-power images at bottom. Arrowheads and arrows indicate p-SMAD2⁻ normal vessels in WT and p-SMAD2/3⁺ vessels in the mutant retinas, respectively. (**k**) SMAD2⁺ ECs and p-SMAD2⁺ microaneurysm were quantified. n = 6 mice per group. Data are means ± SEM. P values are indicated, using unpaired, two-tailed Student's t test. Source data are provided as a Source Data file.

Of clinical significance, the hypovascular phenotype with microaneurysm formation observed in the Tfam, Cox10, or Trx2 resembled that of the retinal microaneurysm in aged human retinas[30]. The AVM phenotypes may also resemble that of human primitive retinal vascular abnormalities such as Coats' disease and Leber's miliary aneurysms[55]. Leber hereditary optic neuropathy (LHON) is a well-known mitochondria-associated inherited disease with vision loss[56]. Whether or not Leber's miliary aneurysms and its severe form Coats' disease are associated with mitochondria need further investigations. Moreover, the phenotype encompassing an incomplete formation of the retinal primary vascular plexus and absence of the deep plexi with a microaneurysm at the tip of the penetration vessels seems to resemble the pathological features of familial exudative vitreoretinopathy (FEVR)[5–10]. FEVR is a human disease characterized by defective retinal angiogenesis and is associated with complications that can result in vision loss. Although defective WNT/β-catenin signaling is an established cause of FEVR[57], EC-specific deletion of serum response factor (SRF) or integrin-linked kinase (ILK) results in sprouting defects, reduced endothelial proliferation, and disruption of the blood-retina barrier, all of which resemble phenotypes seen in established mouse models of FEVR[50,51]. These data suggest that other molecular alterations may also contribute to FEVR disease. Since defective cell-matrix interactions are linked to WNT signaling and FEVR, further investigation is warranted to determine how mitochondrial dysfunction is linked to endothelial WNT, SRF, or ILK signaling in FEVR.

In conclusion, we uncover a novel mechanism whereby mitochondrial dysfunction via the ALK5-SMAD2 signaling induces retinal vascular malformations, and our study has therapeutic value for the alleviation of the angiogenesis-associated human retinal diseases.

## Methods
All data and the methods supporting this study's findings are available from the corresponding authors on request.

### Generation of tamoxifen-inducible endothelial-specific deficient mice (Trx2[ECKO], Tfam[ECKO], COX10[ECKO])
Trx2[fl/fl] mice was generated in our lab as we previously described[22]. Tfam[fl/fl] and Cox10[fl/fl] mice were obtained from Dr. Navdeep Chandel, Northwestern University)[20] and Dr. Carlos Moraes (University of Miami)[21], respectively. Trx2[fl/fl], Tfam[fl/fl] and Cox10[fl/fl] mice crossed with Cdh5-CreER[T2] mice in which the Cre recombinase expression is driven by the Cdh5 promoter to generate mice with inducible endothelial cell (EC)-specific deletion of these three genes. For the in vivo tamoxifen-induced gene deletion, tamoxifen (Sigma, T5648) was diluted at 10 mg/ml in corn oil and fed mice at a dose of 50 μg once daily from postnatal day (P)1 to P3. For pups, littermates were randomly separated into vehicle (WT) and tamoxifen-fed (Tfam2[ECKO], Cox10[ECKO] and Trx2[ECKO] groups such that the sex ratios of mice in both groups were equal. For genetic rescue by Smad2 deficiency, Cdh5CreER[T2]:Tfam[fl/fl] were bred with Cdh5CreER[T2]:SMAD2[fl/fl41] (obtained from collaborator Dr. Anne Eichmann at Yale) to Cdh5CreER[T2]:Tfam[fl/fl]:Smad2[fl/fl]. Smad2[ECKO] and Tfam[ECKO]Smad2[ECKO] (DKO; inducible EC-deletion of Tfam and Smad2) were obtained upon feeding Cdh5CreER[T2]:Smad2[fl/fl] and

Cdh5CreER[T2]:Tfam[fl/fl]:Smad2[fl/fl] pups, respectively, at P1 to P3 with tamoxifen. For adult mice, male and female animals were used in equal numbers for all experiments. Mice were housing under diet from Charles River Laboratory and were housed in the animal care facility of Yale University under standard pathogen-free conditions with a 12 h light/dark schedule and provided with food and water ad libitum, temperature was between 20 and 24 °C and relative humidity between 45 and 65 rH. Mice were cared for in accordance with the National Institutes of Health guidelines, and all procedures were approved by the Yale University Animal Care and Use Committee.

### Treatment with TGFβ inhibitor (LY364947)
TGFβ inhibitor LY364947 (L6293; Sigma-Aldrich, St. Louis, MO, USA) was dissolved in 10% DMSO as 0.5 mg/ml. Littermate of WT, Tfam[ECKO], Cox10[ECKO], Trx2[ECKO]pups were received daily intraperitoneal injection of TGFβ inhibitor (5 μg/g body weight) or 10% DMSO at P2–P6 or P2–P14. Mice were subjected to analyses on P7 or P15.

### Fluorescent staining of whole-mount retinas
Eyeballs were enucleated and fixed in 4% PFA for 2 h on ice. The cornea, lens, sclera and hyaloid vessels were dissected and removed. Retinal samples were blocked with 5% normal donkey serum in PBST (0.3% Triton X-100 in PBS) overnight at 4 °C, followed by incubation with various antibodies diluted 1:50 in blocking solution overnight at 4 °C. After several washes with PBS, retinas were mounted in fluorescent mounting medium. The following first antibodies were used: isolectin B4 (IB4), TFAM, COX10, TRX2, VEGFR3, Endomucin, CD31, Isolectin B4, SMA, collagen IV, ESM1, ERG, p-SMAD2/3. The secondary antibodies were produced in either donkey or goat and targeted against the appropriate species, with conjugation with AlexaFlour 488, 555 or 647 (Invitrogen) (Supplementary Table 4: Antibodies for immunostaining). Pictures were taken with the same exposure and gain settings using a confocal Zeiss Airyscan 880 microscope (Zeiss, Germany). Vascular areas were outlined using NIH Image J software and quantified as the percentage of the total area of the retina analyzed.

### Immunofluorescence analysis for retina sections[58]
Mouse retinas were harvested, fixed with 4% PFA, embedded in OCT and sectioned (5 μm thick). Slides were washed with PBS twice to remove OCT, and 5% donkey serum in 0.3% TritonX-100 in PBS was used for half hour to block non-specific staining and to permeabilize section tissue. Slides were incubated with primary antibodies such as anti-CD31 overnight at 4 °C, then were washed with PBS three times and incubated with secondary antibodies (1:200 to 1:400) at room temperature for one hour. After washed in PBS three more times, slides were mounted with VECTASHIELD Mounting medium with DAPI (Vector Laboratory), and photographed. TUNEL (Roche, catalog 11684795910) staining was performed according to the manufacturer's instructions.

### FITC-dextran perfusion and mtROS detection
FITC-dextran (FD2000S; Sigma-Aldrich, St. Louis, MO, USA) was dissolved in sterile normal saline at a concentration of 50 mg/ml followed

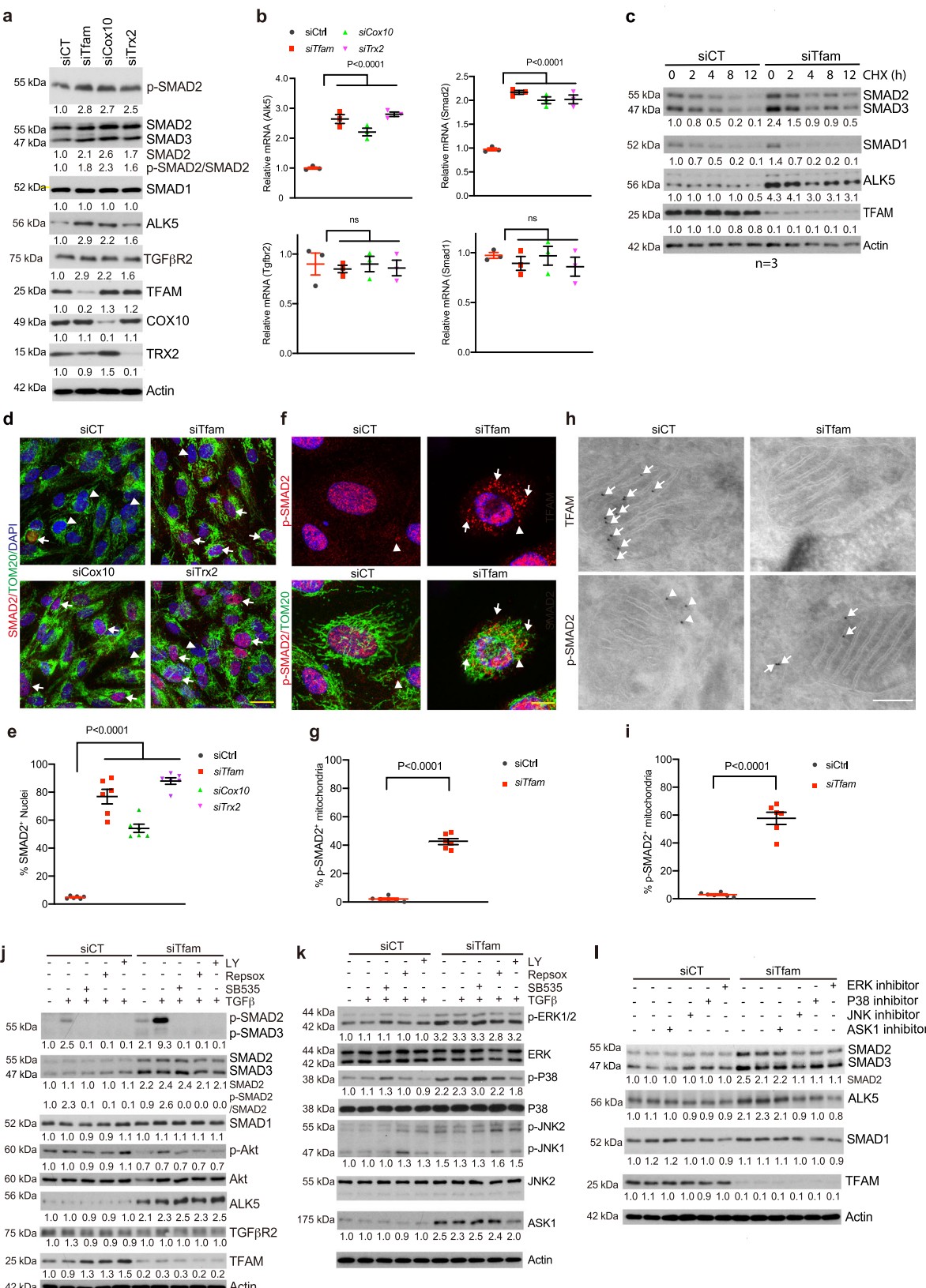

by centrifugation at 10,000 x g for 5 min. The supernatant was collected and protected from light for injection. Littermate of WT, *Tfam2*^ECKO, *Cox10*^ECKO and *Trx2*^ECKO pups were weighed and anesthetized with an intraperitoneal injection of anesthetic (katamine 100 mg/kg + xylazine 10 mg/kg). The retro-orbital injection was followed by the published protocols. Briefly, the lateral canthus of the left orbit was chosen as the injection site. A 31-gauge needle with a 3/10 ml insulin syringe was used to gently pierce into the mouse's orbital venous sinus about 2–3 mm with the bevel of the needle facing upward at a 45o angle. 10 ml per g body weight of FITC-dextran was injected into each pup and waited for 5 min. Fresh retinas were harvested and dissected for cornea, lens, sclera and hyaloid removed. The isolated retinas were

**Fig. 7 | Mitochondrial dysfunction augments TGFβ-ALK5-SMAD2 signaling.**
HUVECs were transfected with control, *Tfam*, *Cox10*, or *Trx2* siRNAs for 48 h. **a** Cell lysates were subjected to Western blotting with respective antibodies. Protein levels were quantified and presented as fold changes by taking control siRNA as 1.0. $n = 3$ (three independent experiments). **b** mRNA expression of *Alk5*, *Tgfbr2*, *Smad*1, and *Smad*2 was determined by qRT-PCR with specific primers. Relative mRNA levels were quantified and presented as fold changes by taking control siRNA as 1.0. $n = 3$ (three independent experiments). **c** Cells were treated with cycloheximide (10 μg/ml) for indicated times. Cell lysates were subjected to Western blotting with respective antibodies. Protein levels were quantified and presented as fold changes by taking control siRNA as 1.0. $n = 3$ (three independent experiments). **d**, **e** Cells were subjected to indirect immunofluorescence staining for SMAD2 or phosphor-SMAD2 and mitochondrial marker TOM20. Arrowheads and arrows (d) indicate SMAD2⁻ and SMAD2⁺ nuclei, respectively. % nuclear SMAD2 was quantified (**e**). **f** Arrowheads and arrows indicate phosphor-SMAD2⁻ and phosphor-SMAD2⁺ mitochondria, respectively. % phosphor-SMAD2⁺ mitochondria were quantified (**g**).

$n = 6$ (duplicates from three independent experiments; 10 cells were counted for each biological sample). **h**, **i** Immuno-EM. HUVECs were transfected with control or *Tfam* siRNAs for 48 h and subjected to immunogold electron microscopy for TFAM and phosphor-SMAD2. Arrows and arrowheads indicate immunogold particles (5 nm) outside and inside mitochondria, respectively. %phosphor-SMAD2⁺ mitochondria were quantified (**h**). 10 EM section per sample. $n = 3$ per group. **j–l** ECs were pre-treated with TGFβR1 inhibitors (5 μM each) for 60 min followed by treatment with TGF-β (10 ng/ml) for 30 min. **l** ECs were pre-treated with LY3214996 (ERK inhibitor), SP600125 (JNK inhibitor), SB203580 (p38 MAPK inhibitor) or GS444217 (ASK1 inhibitor) (5 μM each) for 2 h. Cell lysates were subjected to Western blotting with respective antibodies. Protein levels were quantified and presented as fold changes by taking control siRNA as 1.0. Ratios of p-SMAD2/SMAD2 and p-Akt/Akt are also presented. $n = 3$ (three independent experiments). Data are means ± SEM. $P$ values are indicated, using one-way ANOVA followed by Tukey's multiple comparisons test. Scale bar: 50 μm (**d**); 10 μm (**f**); 150 nm (**h**). Source data are provided as a Source Data file.

subjected to MitoSox staining with MitoSOX™ Red mitochondrial superoxide indicator (M36008; Invitrogen). Mean fluorescence intensity (MFI) of FITC-Dextran and MitoSOX were quantified by ImageJ.

## Transmission electron microscopy (TEM) and electron tomography
Retina tissues were fixed in 2.5% glutaraldehyde and 2% paraformaldehyde in 0.1 M sodium cacodylate buffer pH7.4 containing 2% sucrose for 1 hour and post-fixed in 1% osmium tetroxide for 1 hour. The sample was rinsed in buffer and en-bloc stained in aqueous 2% uranyl acetate for 1 h followed by rinsing in distilled water, dehydrated in an ethanol series and infiltrated with Embed 812 (Electron Microscopy Sciences) resin. The samples were place in silicone molds and baked at 60 °C for 24 h. Hardened blocked were sectioned using a Leica UltraCut UC7. 60 nm sections were collected on formvar coated nickel grids and 250 nm sections on copper slot grids and stained using 2% uranyl acetate and lead citrate. 60 nm sections on grids were viewed FEI Tencai Biotwin TEM at 80Kv. Images were taken using Morada CCD and iTEM (Olympus) cellSens Dimension software. For electron tomography, 250 nm thick sections with 15 nm fiducial gold to aid alignment. Tomography was done using FEI Tecnai TF20 at 200Kv. Data was collected using SerialEM3.8 (Boulder) on a FEI Eagle 4Kx4K CCD camera using tilt angles −60° to 60° and reconstructed using Imod4.9 (University of Colorado Boulder).

## Immunoblotting
Tissue or cells were lysed in 2x Laemmli buffer and boiled for 5 min at 100 °C followed by centrifugation at 20,000x*g* for 15 min at 4 °C. The protein extracts were subjected to standard Western blot analysis which was performed. The following antibodies were used for western blotting: Antibody against COX10 (ab84053), TRX2 (ab185544) from Abcam; p-Akt (rabbit, 9271), Akt (rabbit, 9272), p-SMAD1/5/9 (13820), p-SMAD2/3 (8828), SMAD2/3 (8685), SMAD1 (9743), SMAD2 (5339), TFAM (8076) were from Cell Signaling Technology; TGFβR1 (E-AB-70130; recognizing human and mouse), and TGFβR2 (E-AB-60379) were from Elabscience; ALK1 was from Origen (AP01172PU-N) and ALK5 was from R&D (AF3025-sp; recognizing human); β-actin (mouse, A1978) were from Sigma; p-SMAD2 (44–244 G) and SMAD1 (38-5400) were from Thermo-Fisher. All primary antibodies were diluted 1:1000. For data presented in the same figure panel, the samples were derived from the same experiment and that gels/blots were processed in parallel. Uncropped gel images were provided in Supplementary Fig. 12.

## Gene expression
Total RNA was isolated from tissues or cells using the RNeasy kit with DNase I digestion (Qiagen, Valencia, CA). Reverse transcription was performed using standard procedures (Super Script first-strand synthesis system; Qiagen) with 1 μg of total RNA. Quantitative real-time polymerase chain reaction (qPCR) was performed using iQ SYBR Green Supermix on an iCycler real-time detection system (Bio-Rad Laboratories, Inc., Hercules, CA). qRT-PCR was performed with specific primers (Provided in Supplementary Table 5).

## Cell culture, inhibitors, cytokines, and transfection
Human umbilical vein EC (HUVEC) was from Yale Endothelial Cell Facility (Yale University). Primary mouse retinal endothelial cells were isolated from P7 mouse retinas by the method described previously[59,60]. Briefly, mouse retinas were digested with 1 μg/ml collagenase CLS2 (Worthington) in Dulbecco's modified Eagle's medium (DMEM; Sigma) containing 50 μg/ml gentamycin and 2 mM glutamine in a shaker for 2 h at 37 °C. The cell pallet was separated by centrifugation in 20% bovine serum albumin (BSA)−DMEM (1,000 × g, 20 min). The microvessels obtained in the pellet were further digested with 1 μg/ml Collagenase/Dispase (Roche) in DMEM for 1.5 h at 37 °C. Retinal endothelial cell clusters were separated on a 33% continuous Percoll gradient, and washed twice in DMEM before planting on collagen type IV and human fibronectin-coated 35-mm plastic dishes. Cultures were maintained in DMEM supplemented with 20% FBS and 1 ng/ml basic fibroblast growth factor (Roche). When the cultures reached 80% confluency (5–6 days in vitro), the endothelial cells were passed by brief treatment with trypsin (0.05% wt/vol)−EDTA (0.02% wt/vol) solution (Sigma) for experiments. Mouse retinal EC were confirmed by positive immunostaining for EC markers CD31 (or PECAM-1) and von Willebrand factor (vWF) with negative for astrocyte marker GFAP and pericyte marker NG2.

LY3214996 (ERK inhibitor), SP600125 (JNK inhibitor), SB203580 (p38 MAPK inhibitor) were purchased from EMD Millipore. GS444217 (ASK1 inhibitor) was purchased from Gilead Company.

Human recombinant TGFβ were from R&D Systems Inc. (Minneapolis, MN) and used at 10 ng/ml. siRNAs were purchased from Santa Cruz: si-Tfam (sc-45912), si-Cox10 (sc-72304), si-Trx2 (sc-44173). siRNAs (20 μM) were transfected into cells by Oligofectamine (Life Technologies, Inc.; Invitrogen), following protocols provided by the manufacturer (Invitrogen). At 48 h post-transfection, cells were harvested for protein analysis.

## Mitochondrial function assays
Cellular oxygen consumption was assessed using the Seahorse XF96 analyzer. Briefly, 20,000 ECs were seeded into each well of gelatin-coated XF96 cell culture microplates and cultured in a growth medium. On the day of the experiment, the ECs were washed and then cultured in Agilent Seahorse XF Base Medium

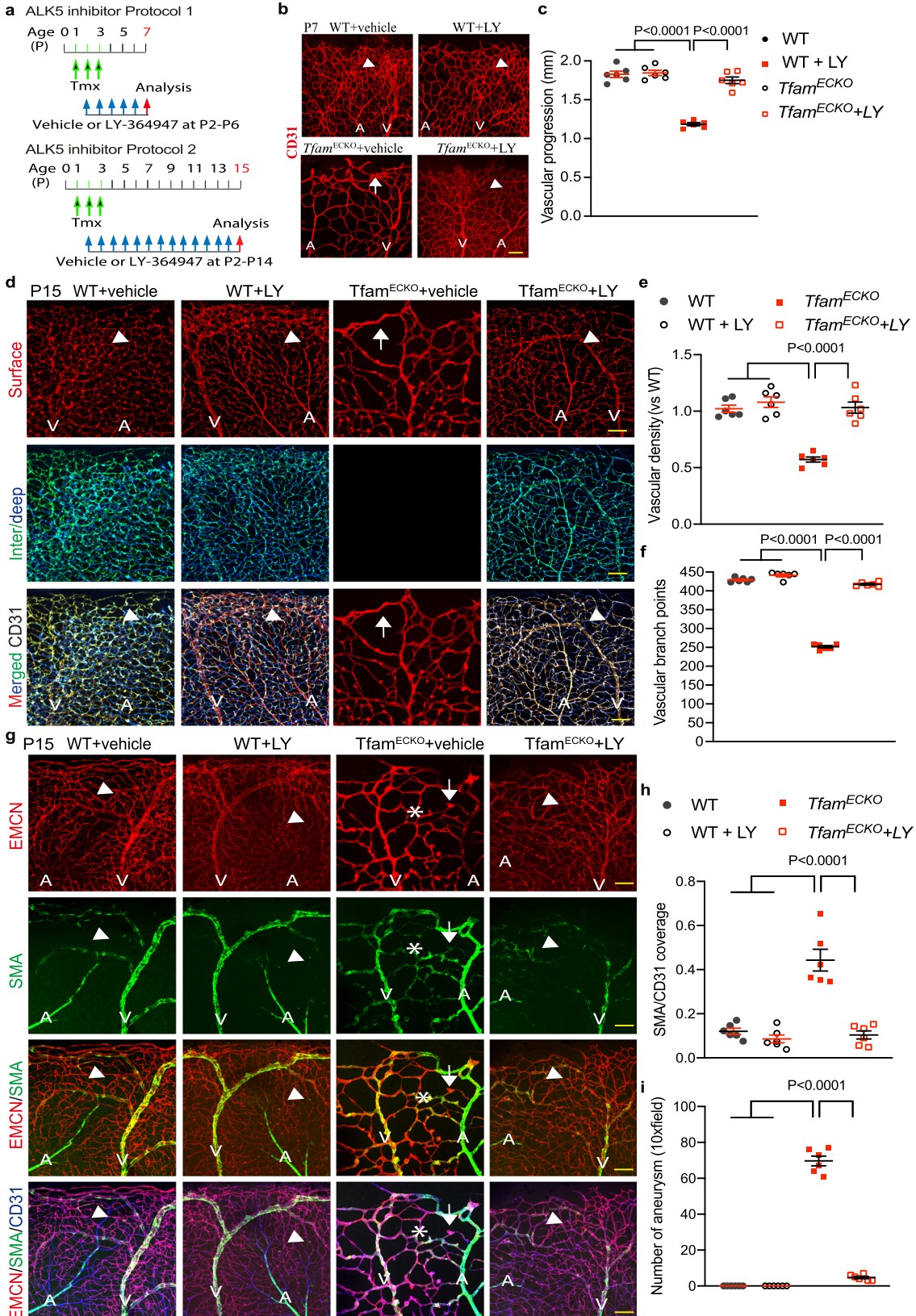

(supplemented with 25 mM glucose and 1 mM pyruvate) in a non-$CO_2$ incubator at 37 °C for 1 h. The cells were then transferred to a temperature-controlled Seahorse analyzer to perform the mitochondrial stress test. After obtaining three basal respiration measurements, we obtained three respiration measurements after adding the coupled respiration inhibitor oligomycin (1 μM) to

assess ATP production and proton leak. We measured maximal respiration three times after adding 1 μM carbonyl cyanide-4-(trifluoromethoxy) phenylhydrazone, and measured spare respiratory capacity and non-mitochondrial respiration after adding 0.5 μM rotenone and 0.5 μM antimycin A. Results were analyzed with Wave software (Agilent Technologies).

**Fig. 8 | ALK5 inhibitor attenuates EC proliferation and the retarded vessel growth in *Tfam*^ECKO retinas. a** A diagram for the ALK5 inhibitor protocols. WT and the mutant pups were received i.p. injection of vehicle or ALK5 inhibitor LY364947 compound at 5 µg/g body weight daily from P2-P6 (Protocol 1) or P2-P14 (Protocol 2). Mice were subjected to analyses at P7 or P15. **b**, **c** Retina lysates were subjected to Western blotting with respective antibodies. Protein levels were quantified and presented as fold changes by vehicle WT as 1.0. Ratios of p-SMAD2:SMAD2 are presented as a graph (**c**). Data presented from one of 3 mice per group. **d**, **e** P7 retinas were subjected to whole mount staining for CD31. Vascular progression (the distance of radial extension of the vascular plexus from the optic center) was quantified (**e**). *n* = 6 mice per group. **f**, **h** Vehicle or LY364947 treated WT and *Tfam*^ECKO mice at P7 were injected with EdU (100 µg per gram of body weight). 4 h after i.p. injection of EdU, retinas were harvested and subjected to whole-mount staining with ERG (red) and CD31 (blue) followed by the EdU Click-iT assay (green). Arrows and arrowheads indicate EdU⁺ and EdU⁻ EC, respectively. EC density (ERG⁺/mm² retinal area) and EC proliferation (EdU⁺ERG⁺/total ERG⁺ ECs) were quantified by taking WT vehicle as 1.0 (**g**, **h**). *n* = 6 mice per group. Data are means ± SEM. *P* values are indicated, using one-way ANOVA followed by Tukey's multiple comparisons test. Scale bar: 100 µm (**b**); 25 µm (**d**, **g**). Source data are provided as a Source Data file.

## Detection of mitochondrial DNA contents and mtROS production

Mitochondrial DNA was measured by real-time-PCR as described previously[22]. To measure mitochondrial ROS (mtROS) production in WAT samples, epididymal WAT was isolated from 14-week-old WT/*Trx2*^ADKO mice, placed in OCT compound (without fixation), and frozen. The fresh frozen 30-µm sections were washed in PBS for 15 min and then incubated with 2.5 µM MitoSOX (Molecular Probes; M36008) at 37 °C for 10 min followed by detection under microscope with the emission/excitation at 580 / 510 nm. The sections were washed again with PBS, stained with DAPI, and visualized using a fluorescent microscope (Zeiss). Fluorescence intensity were quantified in an average of six random fields per section. To determine mtROS production in differentiated adipocytes (day 8), adherent cells were washed with PBS three times and then incubated in 2.5 µM MitoSOX at 37 °C for 10 min. Images were acquired using a confocal microscope (Zeiss).

## Isolation of mitochondria and assessment of mitochondrial function

Mitochondria from ECs were isolated using a mitochondria isolation kit for tissue (Thermo Scientific; 89874) according to the manufacturer's instructions. To determine the amount of ATP generated by the mitochondria, the cell pellet containing mitochondria was resuspended in sterile distilled water and assessed with an ATP bioluminescent assay kit (Sigma Aldrich; FLAA) following the manufacturer's protocol.

For protein analyses, the cytosolic fraction and pellet containing mitochondria were boiled with Laemmli buffer, loaded on an SDS-PAGE gel and analyzed by Western blotting.

## EdU incorporation assays

For EdU incorporation assay in vitro, cells were incubated with 10 µM 5-ethynyl-2′-deoxyuridine (EdU) for 4 h prior to fixation with 4% PFA for 30 min at room temperature. Once fixed, cells were subjected to EdU assays using the Click-iT EdU kits (ThermoFisher Scientific).

For in vivo EdU incorporation assay, 100 µg EdU (Sigma) per gram of body weight dissolved in sterile PBS was injected intraperitoneally into the pups. 4 h after EdU injection, retinas were harvested and subjected to EdU assays using the Click-iT EdU kits (ThermoFisher Scientific).

## A transwell migration assay

Cell migration was examined using Transwell fitted with polycarbonate filters (8-µm pore size) (Corning Inc.). Briefly, the under surface of the filters were coated overnight at 4 °C with fibronectin (20 µg/ml) in PBS (pH 7.4). The coating solution was removed from the lower chamber before filling with migration medium. The lower chambers were filled with DMEM containing 0.5% BSA, with or without VEGF (50 ng/ml). Cells were harvested in PBS containing 5 mM EDTA, and after washing with DMEM containing 5 mM MgCl₂, cells were resuspended to a concentration of 1 × 10⁶ cells/ml DMEM containing 0.5% BSA. We loaded 1 × 10⁵ cells (0.1 ml) into each upper chamber and cells were

cultured for 6 h at 37 °C in the presence of 5% CO₂ in a humidified incubator. Cells that did not migrate through the coated filters were removed with cotton swabs, and cells that migrated to the lower surfaces of the filters were stained for 30 min with 0.2% crystal violet in 10% ethanol. Cells on the lower surfaces of the filters were imaged under microscope and quantified.

## Three-dimensional bead sprouting assay

HUVECs were transfected with siRNAs overnight by Lipofectamine RNAiMAX following protocols provided by the manufacturer (Invitrogen) and changed to fresh prepared Microvascular Endothelial Cell Growth Medium-2 MV (EGM2, Lonza). Three-dimensional bead sprouting assay was performed according to a published protocol[58,61]. Briefly, cells were harvested at 24 h post-transfection and were coated with Cytodex 3 microcarrier beads (C3275, Sigma) with a concentration of 400 cells per bead in EGM2 medium. These coated beads were embedded in 2 mg/ml fibrin gels in 24-well plates by mixing 2 mg/ml fibrinogen (Calbiochem) in DPBS, 0.625 U/mL thrombin (Sigma-Aldrich), and 0.15 Units/ml aprotinin (Sigma-Aldrich). EGM2 medium containing fibroblasts (20,000 per well) was added to each well. The cells were cultured and maintained for 2–8 days by changing the medium every other day. Bright-field images were captured with Axiovert 200 (Zeiss) at ×10 magnification and sprout lengths were measured with NIH Image J.

## Cytotoxicity assay

A LIVE/DEAD viability/cytotoxicity assay Kit (ThermoFisher, L3224) was performed. The commercial kit consists of a two-color fluorescence cell viability assay with two probes that recognize intracellular esterase activity (calcein AM) and plasma membrane integrity (ethidium homodimer) and consequently produce green fluorescence in live cells and red in dead cells. Reagents were used according to manufacturer specifications. Fluorescent images were obtained using Axiovert 200 (Zeiss) at 10X magnification immediately upon completion of the assay.

### scRNA-seq analyses

**Methods for single-cell FACS sorting.** Retinal tissues (n = 3 mice for each group) were sliced into small fragments roughly 6–10 mm³. The finely minced tissue is transferred to a digestion mix consisting of DMEM + 1 mg/ml collagenase type A + 0.5 mg/ml dispase for 1 h at 37 °C and pipetting every 10 min. DAPI is used to detect dead cells. The cell suspension is passed through a 100 µm filter before sorting. A FACS machine is used to sort CD31⁺CD45⁻ live ECs. Single cells are sorted into 0.04% BSA/PBS with 5000–10,000 cells are sufficient.

**Methods for droplet-based scRNA-seq library preparation and sequencing.** Sorted cells were processed for scRNA library preparation using the Chromium™ Single Cell Platform (10x Genomics) as per the manufacturer's protocol. Briefly, single-cell suspensions were partitioned into Gel Beads in Emulsion in the Chromium™ system (10x Genomics), followed by cell lysis, and barcoded reverse transcription of RNA and cDNA amplification. Final scRNA-seq libraries were sequenced on an Illumina NextSeq 6000.

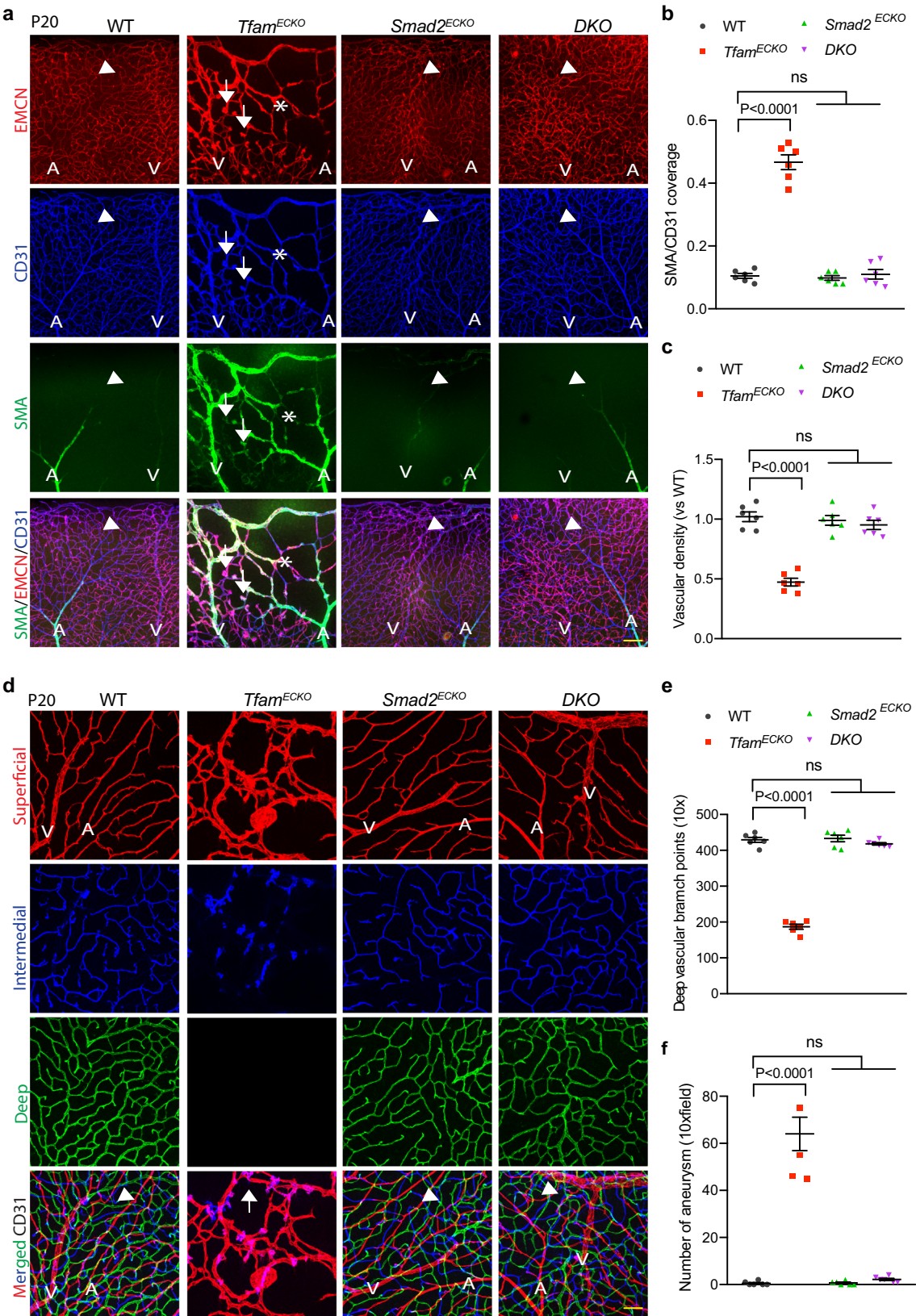

**Data processing of scRNA-seq.** Alignment, filtering, barcode counting, and unique molecular identifier (UMI) quantification for determining gene transcript counts per cell (generating a gene-barcode matrix) were performed by the count function of Cell Ranger (v3.0.2; 10x Genomics) with default parameters and the mouse reference genome, mm10 (Ensembl build 93). Filtered feature-barcode matrices for each library from the Cell Ranger workflow were further analyzed using the Seurat R package (v.4.1.1)[62]. Cells and genes were filtered based on quality following standard procedures. Cells with a unique feature count between 500 and 6000 and with less than 20%

**Fig. 9 | Smad2 genetic deficiency rescues the vascular malformation in *Tfam*^ECKO mice.** Retinal tissues from WT, *Tfam*^ECKO, *Smad2*^ECKO and double knockout (DKO) mice were harvested at P15. **a−c** Attenuation of arterialization and microaneurysm formation in the DKO retinas. Whole mount co-staining with α-SMA, endomucin (EMCN) and CD31. Arrowheads indicate normal or normalized vessels and arrows indicate microaneurysm in the mutant retinas. Asterisks indicate arterialized microvessels. α-SMA coverage (α-SMA/CD31 ratio) and vascular density were quantified. *n* = 6 mice per group. A: artery/arterial; V: vein. **d−f** Whole mount

staining for CD31. Vasculatures in three retinal layers (surface, intermediate and deep) were imaged by confocal microscopy analyses, and pseudo-colored by red, blue and green, respectively. *Tfam*^ECKO retinas had only the surface vessels with malformation and microaneurysms (arrows). Normal (WT and *Smad2*^ECKO) or normalized capillaries in DKO are indicated by arrowheads. Vessel density and branch points were quantified. n = 6 mice per group. Data are means ± SEM. *P* values are indicated, using one-way ANOVA followed by Tukey's multiple comparisons test. Scale bar: 50 μm (**a**); 25 μm (**d**). Source data are provided as a Source Data file.

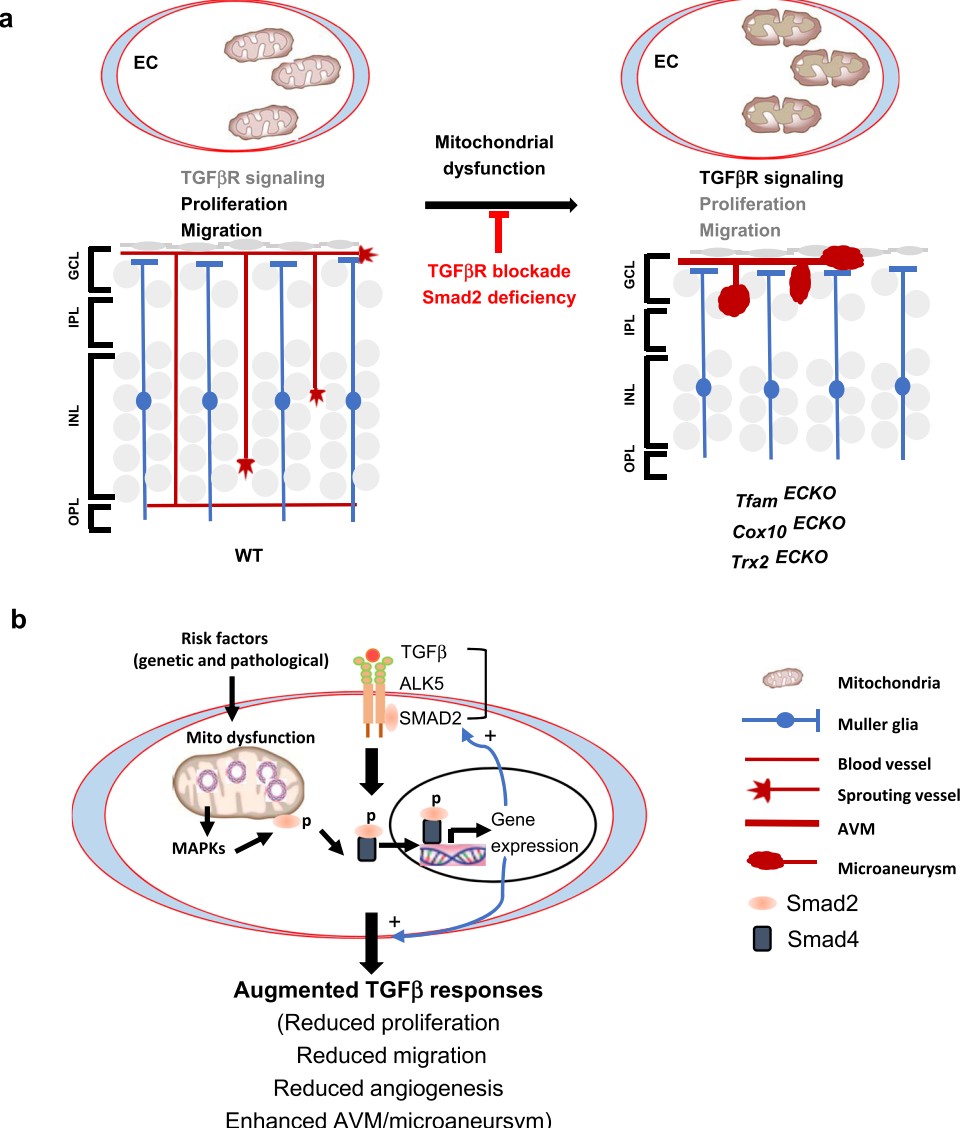

**Fig. 10 | A proposed model for TGFβ signaling mediates mitochondrial dysfunction-induced vascular malformation. a** A summary for mitochondrial dysfunction-induced vascular malformation. Normal retinal vessels begin to dive into the deep layers around P8 to form three layers of vascular network around P12-21 (the surface ganglier layer, the intermediate inner plexus layer and the deep outer plexus layer). Retinas in mice with an inducible deletion of *Tfam*, *Cox10*, or *Trx2* exhibit retarded vessel growth without penetration into the deep plexi, and developed arteriovenous malformations (AVM) with microaneurysm formation. While tip EC retains a low level of TGFβ signaling with a high proliferation and migratory

activity, the overactivated ALK5-SMAD2 signaling mediates both retarded vascular growth and malformations. Pharmacological blockade of ALK5 or genetic deletion of *Smad2* prevent the retinal vascular malformation. **b** Mechanism for mitochondrial dysfunction-induced TGFβ signaling. We propose that that mitochondrial dysfunction induces activation of the MAPK pathway, leading to enhanced transcription of TGFβ signaling components (including TGFβ, ALK5 and SMAD2); increased TGFβ signaling can in turn attenuate mitochondrial function, forming feedback loop (see text for details).

mitochondria content, and genes expressed in more than three cells were retained. In addition, to further enrich for ECs, only cells expressing *Pecam1* and not *Ptprc*, were kept for subsequent analyses. Normalization and variance stabilization was performed with Seurat SCTransform function (v2)[63] with the glmGamPoi package[64]. The

RunUMAP function with 30 dimensions in its default setting was applied to obtain the UMAP projections. The FindClusters function with a 0.8 resolution was carried out to cluster cells into different groups. Cell clusters were annotated into EC subtypes based on visual inspection (Supplementary table 1) and GO analyses (Supplementary

Table 2) of the positive differentially expressed genes for each cluster compared to all other cells. To identify differential expression genes between WT and the different mutants within ECs, the Seurat Find-Markers function was applied with the default parameter using a Wilcoxon Rank Sum test and excluding cells from clusters 8 and 9 which were identified as pericytes cells and neuron cells. Genes were considered significantly differentially expressed when the adjusted *p*-value, based on Bonferroni correction using the total number of genes in the dataset, was below 0.05.

Gene Ontology analysis was performed on the DEGs by GOstats Bioconductor package (v2.46.0)[65]. False discovery rate (FDR) was adjusted by p.adjust function in R with "method = "BH"" parameter.

### Statistics and reproducibility
Group sizes were determined by an a priori power analysis for a two-tailed, two-sample Student's *t* test with an α of 0.05 and power of 0.8, in order to detect a 10% difference in lesion size at the endpoint. Animal were grouped with no blinding but randomized during the experiments. Male and female animals were used in equal numbers for all experiments. No samples or animals were excluded from analysis. All quantifications (lesion sizes, junctional integrity, sprout length and lumen) were performed in a blind fashion. All figures are representative of at least three experiments unless otherwise noted. All graphs report mean ± standard error of mean (SEM) values of biological replicates. The normality and variance were not tested to determine whether the applied parametric tests were appropriate. Comparisons between two groups were performed by unpaired, two-tailed Student's *t* test, between more than two groups by one-way ANOVA followed by Tukey's multiple comparisons test or by two-way ANOVA followed by Sidak's multiple comparisons test using Prism 6.0 and 8.0 software software (GraphPad). *P* values were two-tailed and values <0.05 were considered to indicate statistical significance. Exact *p*-values and the respective test/analysis are listed in the figure and legends.

### Reporting summary
Further information on research design is available in the Nature Portfolio Reporting Summary linked to this article.

## Data availability
All data, including data associated with main figures and supplementary figures, are available within the article or in the online-only Data Supplement or from the corresponding author on request. Source data are provided with this paper.

Accession codes, unique identifiers, or web links for publicly available datasets: scRNA-seq data have been deposited and accession codes are provided. The scRNA-seq data have been uploaded to the Gene Expression Omnibus (GEO) data repository (http://www.ncbi.nlm.nih.gov/projects/geo/) under accession number GSE172230.

Source data are provided with this paper. A list of figures that have associated raw data: Main figures (Figs. 1–9), Supplementary tables (Tables 1–3) and Supplementary figures (Figs. 1–10) associated with raw data are presented in the source data.

A description of any restrictions on data availability: No restrictions on data availability. Source data are provided with this paper.

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

## Acknowledgements

This work was partly supported by NIH grants US National Institutes of Health (NIH) grants R01EY033333 and HL157019 to J.H.Z. We appreciate Georgia Zarkada's help on the Smad2/3 flox mice, and Al Mennone's help in acquiring images at the Yale University Center for Cellular and Molecular Imaging (CCMI); We appreciate Professor Jordan S. Pober for his comments. The sequencing service was conducted at the Yale Stem Cell Center Genomics Core facility and was supported by the Connecticut Regenerative Medicine Research Fund.

## Author contributions

J.H.Z. and W.M. provided conceptual of the study and designed all the experiments; J.H.Z. executed and organized team members; H.Z. for mouse breeding, genotyping, and Western blotting; B.L., J.H.Z., Q.H., and Q.L. for in vivo sample staining; J.H.Z. for 3D angiogenesis assays; Y.T., S.M., I.H.P. and G.W. for bioinformatic analyses; M.G. performed EM guided by XL; A.E. for editing and discussion; J.H.Z. and W.M. analyzed the data, wrote the manuscript and provided funding sources.

## Competing interests

The authors declare no competing interests.

## Additional information

[1]Interdepartmental Program in Vascular Biology and Therapeutics, Department of Pathology, Yale University School of Medicine, New Haven, CT, USA. [2]Yale Center for Genomic Analysis, Department of Genetics, Yale University School of Medicine, New Haven, CT, USA. [3]Yale Stem Cell Center, Department of Genetics, Yale University School of Medicine, New Haven, CT, USA. [4]Maisonneuve-Rosemont Hospital Research Center (CRHMR), Department of Medicine, University of Montreal, Montreal, QC, Canada. [5]Department of Cell Biology, Yale University School of Medicine, New Haven, CT, USA. [6]Department of Internal Medicine (Cardiology) and Cellular and Molecular Physiology, Yale University School of Medicine, New Haven, CT, USA. [7]These authors contributed equally: Haifeng Zhang, Busu Li, Qunhua Huang. ✉e-mail: mike.wang388@gmail.com; Jenny.zhou@yale.edu

