## [Peer Review File · Nature Communications]

Reviewers' Comments:

Reviewer #1:

Remarks to the Author:

Zhang and colleagues focus on the role of mitochondria (and in particular of mitochondrial activity) in angiogenesis. To do that, they use three genetic independent approaches which allow to target (nuclear encoded) mitochondrial proteins with distinct functions and combine in vitro, in vivo and scRNAseq analysis (promoted by the deletion of Tfam, Cox10, or Trx2). These combined strategies allowed to identify that Tfam, Cox10, or Trx2 depletion results in the appearance of AVMs and microaneurysms. Mechanistically, they identify that disruption of mitochondrial activity enhances the TGFb-ALK5-SMAD2 axis. Also, pharmacological inhibition of ALK5 prevents AVM in the Tfam, Cox10, or Trx2.

This manuscript is experimentally well performed, and the data are convincing. The findings are relevant and may have important biological and therapeutic implications. This study would benefit by clarify some points as described below.

1- Authors have identified an interesting role/function of mitochondrial activity in endothelial pathophysiology in mice. The key question is whether this is also relevant in vascular malformations in humans. Authors suggest that the phenotype identified resemble that that is observed in Coats' disease, Leber's military aneurysms and familial exudative vitreoretinopathy, and human familial exudative vitreoretinopathy (FEVR). This manuscript would benefit from a better link between the phenotype and these diseases.

2- Authors identify that mitochondrial activity plays a role during angiogenesis by targeting 3 mitochondrial proteins with distinct functions (mitochondrial TF, respiratory complex IV component, and a mitochondrial redox protein). So how come targeting the mitochondrial signaling vs. mitochondrial power function results in a similar phenotype? This is surprising, as the current trend is that mitochondria mainly function as signalling organelles during this physiological process, and not as energy source. Can the authors elaborate more on this?

3- In line with the previous point, a better characterization (beyond a phenotype that papers by disrupting mitochondrial genes) of mitochondria during angiogenesis should be included. For instance, assessment and state of mitochondrial content in ECs during angiogenesis, and the impact of mitochondria upon depletion of Tfam, Cox10, or Trx2. Authors indicate that mitochondria are damage in Tfam, Cox10, or Trx2 (Fig. 3i,j), what does this mean? An important observation of this manuscript is that sprouting into deeper layers seems to be severely affected upon depletion of Tfam, Cox10, or Trx2. Why is that?

4- What is the impact of deletion of Tfam, Cox10, or Trx2 on cell death? I am wondering if the observation that all mutants result in an overall shut down of important biological functions critical for the expansion of the vasculature is simply a consequence of a massive cellular collapse.

5- The arterialization of the vasculature should be validated by using intrinsic endothelial (arterial) cell markers such as Sox17 or Cx40.

6- scRNAseq is an elegantly approach which aims to provide some mechanistic understanding. This analysis would benefit by providing a better lay out of what it is different and what it is similar between the three models used.

7- It would be interesting to validate whether EndMT occurs in vivo.

8- The TGFb pathway has been extensively involved in the appearance of AVM. In this study, author identify a new family member (ALK5) in the involvement of mitochondrial-related AVM phenotype. However, the phenotype here relates to ALK5 gain of signalling instead of loss of TGFb function/signalling as it occurs in HHT. Is Alk5 a direct target of mitochondrial function or a

consequence of the phenotype? Are the AVM and microaneurysms phenotypes reproduced by using an Alk5 GOF approach?

9- The manuscript is too lengthy (word wise) providing too many unnecessary details. Some of the parts could be simply improved by a making an effort with the wording.

10- Supp Fig 2b requires some revision, i.e. the CD31 and COX10/CD31 and TRF2/CD31 have to swapped positions, the TFAM stained images has some stars and numbers on top.

11- Figures 2g (lower panel) do not have enough resolution.

Reviewer #2:

Remarks to the Author:

In this extensive study the authors examined the effects of altering mitochondrial function on endothelial cell (EC) properties in culture and on vascular anatomy and TGF beta signaling of the developing retina in postnatal mice. In cultured EC's, the authors used siRNA technology to reduce expression of three nuclear genes that contribute to mitochondrial function:

1. Tfam (mitochondrial transcription factor A), which is essential for mtDNA replication, is involved in mtDNA storage (in nucleoids) and is a component of mtDNA transcription. There may also be other functions not yet extensively characterized.
 2. Cox10 (cytochrome oxidase subunit 10) a subunit of Cox involved in both its assembly and electron transport function.
 3. Trx2 (thioredoxin 2) a component of the oxidative stress defense system of mitochondria.
- In mice, the authors generated tamoxifen-driven expression of Cre recombinase in mice that had floxed genes for Tfam, or Cox10, or Trx2. These mice were fed tamoxifen for several days postpartum, then used in experiments. It would have been more elegant to use mice with each gene combination, not given tamoxifen, as CTL's for each mitochondrial gene removed, but this is not worth quibbling about. All of these systems are "leaky" to some degree.

This is an important study that contributes significantly to our understanding of mitochondrial involvement in retinal angiogenesis. As far as I am aware, it is unique (note my lack of expertise in retinal vascular biology). The conclusions appear justified by the data presented. I detect no ethical compromises. The statistical analyses are straightforward.

I have the following comments:

Major comments:

1. This is a large study, part of which is not appropriate for a member of the Nature family journals directed at a general scientific audience. The section on postnatal developmental retinal vascular biology (mainly anatomy) is apparently well done, but very specialized. This material needs to be reviewed by someone with expertise in retinal vascular biology (certainly not me!) and published separately. I'll leave a decision about this point to the Editors.
2. The authors misused the "Supplemental Figures" section as an overflow presentation of many critical findings. I received a separate file of Supplemental figures, so it was relatively easy to read the manuscript and go back and forth from "regular" Figures to Supplemental Figures. I cannot envision a reader trying to navigate the manuscript easily and go back and forth as I did. The authors need to decide what is critical visual information, needing to be in the "regular" Figures, and what is truly "Supplemental" and not needed to prove the main points in the paper.
3. I thought the material on mitochondrial dysfunction was well presented. While one can argue endlessly about the specificity (or lack thereof) of siRNA approaches (or even gene removal), these are established procedures.
4. I also liked the section on single-cell RNAseq, in which EC's from the tamoxifen-treated mice were enriched by flow cytometry, subjected to RNAseq and data reduced by a UMAP algorithm and clustered.
5. The Discussion section is too long, in keeping with the paper being too large and unwieldy. It should be half as long at most, which will easily result if the paper is broken up.

6. Why are there no electroretinograms? A physiological correlate of the mitochondrial dysfunction (and vascular maldevelopment) would be contributory. It would seem to be helpful if ERG's were carried out in the absence and presence of treatment with the TGF beta R inhibitor.

Minor comments:

1. There are several minor grammatical errors and mis-spellings. These are easily corrected.
2. In at least two places the manuscript refers to "Leber's 'military' aneurysms", when the authors meant to say "Leber's military aneurysms". "DISCUSSION" is also mis-spelled.

Reviewer #3:

Remarks to the Author:

The current study by Haifeng Zhang et al., describes a so far unknown molecular mechanism which is responsible for the vascular malfunction upon mitochondrial respiratory inactivation. Two independent recent studies by Diebold et al (Ref#15) and Schiffmann et al (Ref#9) showed that mitochondrial respiratory inactivation by conditionally knocking out the components of complex III (QPC) or complex IV (Cox10), respectively, in endothelial cells (ECs) cause vascular alteration in retinal vascularization (Diebold et al), embryonic development, Wound Healing (Schiffmann et al), and tumour vascularization (Diebold et al and Schiffmann et al). High throughput metabolomics in these studies suggested the cellular metabolic alteration as the underlying molecular mechanism for altered vascular function.

The current study utilizes 3 independent inducible genetic mouse models including TfamECKO, Cox10ECKO, and Trx2ECKO and provides extensive imaging analyses combined with some in vitro investigations showing that mitochondrial respiration in ECs is important for endothelial functions (Figure 1-4). The scRNA-seq analyses (Figure 5-6) identified a number of gene clusters that were altered upon mitochondrial dysfunction. In particular, a large number of TGFβ-responsive genes were altered upon mitochondrial dysfunction (either up- or down-regulated). Finally, by using TGFβR inhibitor (LY-364947 compound at 5 μg/g body weight daily from P2-P6) authors showed that TGFβR inhibition rescued the retarded vessel growth and vascular malformation in the mutant mice.

In my view, the current study is very-well designed and conducted. One important issue is the central conclusion which represents TGFβR signaling as the major molecular mechanism causing vascular malfunction upon mitochondrial respiratory inactivation. This is mainly based on the use of a TGFβR inhibitor. It is not clear how mitochondrial dysfunction involves or interferes with TGFβ-ALK5-SMAD2 signaling? Does TGFβR inhibitor also rescue ECs dysfunction in vitro (Figure 1)? Is the response to TGFβR inhibitor similar in all 3 independent mutant mice or ECs in vitro? Does an increased TGFβ-ALK5-SMAD2 signaling under other experimental conditions cause the same phenotype?

Here are addition points:

In the Introduction the description of the previous works dealing with the role of mitochondrial respiration in ECs is somehow inadequate. Authors stated:

However, recent data suggest that mitochondrial activity is critical for the biosynthetic and bioenergetic demands that are required for angiogenesis. Specifically, mitochondrial respiration, by providing the building blocks and metabolites necessary for EC proliferation rather than producing ATP, regulates angiogenesis at various steps 8,9.

Schiffmann et al showed that mitochondrial respiration is important for EC growth and neoangiogenesis in embryonic development, wound healing and tumour growth. They use cox10 KO mice for their study (similar to the current work).

Its regulatory role was concluded via the effects observed following the inhibition of the mitochondrial glucose metabolizing enzyme pyruvate dehydrogenase (PDHA1), the genetic deletion of fatty acid metabolizing enzyme mitochondrial carnitine palmitoyltransferase 1 (Cpt1), and glutamine limitation in 3D sprouting assays^{7,8,10-12}. More recently, a report showed that the loss of a subunit in complex III (QPC) caused a reduction in EC respiration that impaired cell

proliferation and angiogenesis in mice¹³⁻¹⁵.

The cited works Chen et al¹³ and Diaz et al¹⁴ showed that Complex III alteration causes ROS production. These studies did not investigate any aspects of mitochondrial respiration in ECs. Instead, Schiffman et al⁹ showed that EC-specific COX10 deficiency causes vascular phenotype and must be cited here rather than in the previous position. Coutelle et al (EMBO MO Med 2014, PMID: 24648500) also showed that mice harbouring defect in the proofreading function of the catalytic subunit of mitochondrial DNA (mtDNA) polymerase (POLGA) have an altered neoangiogenesis which could also be cited here as one of the initial studies using genetic mouse model to study the role of mitochondrial respiration in vascularization.

The phenotype of Tfam⁻, Cox10⁻, and Trx2-deficient mice have been characterized in various tissues and cell types; however, little is known regarding their impacts on physiological angiogenesis.

Schiffmann et al used two independent Cre lines and showed that Cox10⁻ in ECs is important for EC growth and neoangiogenesis in embryonic development, wound healing and tumour models

Treatment with LY in TfamEcko mice reduced p-SMAD2/3 levels in ECs derived from these mice (Fig 7b). Increased p-SMAD2/3 levels were detected in siTfam ECs (Figure 6d,e, g) but appeared to be rather unchanged in siCox10 ECs (Figure 6 g) and were not further investigated. Then, how do the authors explain, that TGFβR inhibition also rescues the phenotype in Cox10Ecko mice?

Reviewer #4:

Remarks to the Author:

The authors sorted mouse P12 retinal CD31+CD45⁻ ECs from WT, TfamECKO, Cox10ECKO, and Trx2ECKO mouse models and achieved a scRNA-seq dataset of 4,392 single cells. Then, the authors performed a series of commonly used scRNA-seq data analyses using several R packages, such as Seurat, GOSTATS and destiny. The authors demonstrated heterogeneous EC clusters by UMAP plots, selected GO terms significantly up/down-regulated in mutant mouse models by dotplots and expression of selected signature genes (Actin, ECM, Angiogenic, etc.) by heatmaps, trying to reveal the underlying mechanisms of AVM phenotype in three mutant mouse models. However, description of some results is confusing and needs to be stated more clearly.

Major concerns:

- 1) For the scRNA-seq dataset, submission to a community-endorsed, public repository is mandatory. Accession numbers must be provided in the paper. (According to the Editorial policies of Nature portfolio: <https://www.nature.com/nature-portfolio/editorial-policies/reporting-standards#availability-of-data>). Moreover, without more detailed information (eg. gene expression profiling and metadata for single cells), I cannot make an objective and accurate judgment on some results shown here.
- 2) GO terms shown in Fig. 5c are specifically selected or top most enriched by p values? Please specify it. If not top most enriched, consider to show them in the supplementary data.
- 3) Why did the authors switch from Seurat to destiny to further cluster ECs into nine subtypes? Are there any special considerations? How the batch effect (if it exists) was mitigated using destiny? As far as I know, diffusion map is usually used for Diffusion Pseudo Time analysis, and it is not common to be used for clustering. Please write more details of the clustering procedure in Methods. Besides, the authors stated nine EC subtypes were identified, but there is no further data to be shown to support their heterogeneity and characteristics of each subtype. It could be more informative to exhibit their DEGs (e.g. top 5 or 10) and/or GO terms in the Supplementary Figures or Tables.
- 4) Heatmaps displayed 1) average expression or fold change? 2) value of all ECs in one mouse model or only ECs included in the corresponding subtype (eg. Actin genes in Fig. 5e for Cluster 1)? Please describe these more clearly. If all ECs, whether distribution differences of different mouse models will bias the results? For example, Ribosome genes were highly expressed in three mutant models compared to WT, given that less ECs were in cluster 7 in WT. If not, please state more clearly in the text. Similarly, there are same concerns for Fig. 6a-c and Supplementary Fig. 6b-d. In addition, it seems that Fig. 5e-i were cited incorrectly in the main text (Page 13).
- 5) I noticed that nine EC subtypes might be differentially distributed in four mouse models (Fig.

5d). This would be clearer to show a bar plot of distributions of EC subtypes in each mouse model. Whether the differences in distributions of EC subtypes are also associated with the phenotype of mutant mouse?

6) The authors only showed Fold Changes in the heatmaps (Fig. 5e-i, Fig. 6a-c and Supplementary Fig. 6b-d). Statistical significance should also be shown, and asterisk symbols indicated in each tile is recommended.

POINT-TO-POINT RESPONSE TO REVIEWERS' COMMENTS

Response to Reviewer #1:

General comment: This manuscript is experimentally well performed, and the data are convincing. The findings are relevant and may have important biological and therapeutic implications. This study would benefit by clarify some points as described below.
-We appreciate your positive comments and your instructive suggestions.

1- Authors have identified an interesting role/function of mitochondrial activity in endothelial pathophysiology in mice. The key question is whether this is also relevant in vascular malformations in humans. Authors suggest that the phenotype identified resemble that that is observed in Coats' disease, Leber's military aneurysms and familial exudative vitreoretinopathy, and human familial exudative vitreoretinopathy (FEVR). This manuscript would benefit from a better link between the phenotype and these diseases.

- We have emphasized that the phenotypes observed in our mice that mitochondrial dysfunction induces microaneurysm and basement thickening seems to represent a common feature of human retinal disease, including Leber's miliary aneurysm and its severe form Coats' disease, familial exudative vitreoretinopathy, and human familial exudative vitreoretinopathy (FEVR), and age-related macular degeneration (AMD). Further human genetic and tissue analyses are required for better link of mitochondrial dysfunction-induced phenotype to human disease settings in the future investigations.

2- Authors identify that mitochondrial activity plays a role during angiogenesis by targeting 3 mitochondrial proteins with distinct functions (mitochondrial TF, respiratory complex IV component, and a mitochondrial redox protein). So how come targeting the mitochondrial signaling vs. mitochondrial power function results in a similar phenotype? This is surprising, as the current trend is that mitochondria mainly function as signalling organelles during this physiological process, and not as energy source. Can the authors elaborate more on this?

- We agree with you that mitochondria mainly function as signalling organelles during this physiological process, and not as energy source. Deletion of one of three genes causes not only defects in ATP production, but also induces changes of common gene signatures. Moreover, mitochondrial dysfunction represents a common signaling hub to activate downstream MAPK-SMAD2 signaling, leading to similar phenotypes in the three mutant mice.

3- In line with the previous point, a better characterization (beyond a phenotype that papers by disrupting mitochondrial genes) of mitochondria during angiogenesis should be included. For instance, assessment and state of mitochondrial content in ECs during angiogenesis, and the impact of mitochondria upon depletion of Tfam, Cox10, or Trx2. Authors indicate that mitochondrial are damage in Tfam, Cox10, or Trx2 (Fig. 3i,j), what does this mean? An important observation of this manuscript is that sprouting into deeper layers seems to be severely affected upon depletion of Tfam, Cox10, or Trx2. Why is that?

- We measured DNA copy and ATP production in sorted retinal ECs. Similar to the in vitro results, mitochondrial DNA copy was reduced in isolated retinal ECs from *Tfam*^{ECKO} mice, but not in *Cox10*- or *Trx2*-deficient retinal ECs. However, ATP contents were reduced in all mutant ECs. We also assessed the mitochondrial number and morphology in ECs of retinal tissues by EM. Consistent with the functional alterations, EM images showed mitochondria had reduced cristae without detectable autophagy, apoptosis or membrane damages (Fig.4m-n), unlike that we observed in *Trx2*-deficient cardiomyocytes and adipocytes mitochondria¹⁻³.

Recently, a study from our co-author Dr. Anne Eichmann's group has identified the diving tip cell (D-tip) responsible for diving into the deeper layers⁴. Interestingly, while classical sprouting tip cell (S-Tip) genes were upregulated, the diving tip cell (D-Tip) genes were specifically downregulated in the mutant retinas (Fig. 5d).

4- What is the impact of deletion of *Tfam*, *Cox10*, or *Trx2* on cell death? I am wondering if the observation that all mutants result in an overall shut down of important biological functions critical for the expansion of the vasculature is simply a consequence of a massive cellular collapse.

- EM image analyses revealed altered mitochondrial numbers and structures. Total number of mitochondria were increased in the mutant retinal ECs (**Fig. 4h-i,k**). However, mitochondria in retinal ECs of the mutant mice had disrupted cristae with reduced ratio of cristae surface area/outer membrane surface area, indicative of mitochondrial dysfunction in these mutant ECs (**Fig. 4h-i,l**). There were no obvious apoptotic ECs characterized by blebbing nuclei or mitophagy characterized by typical autophagic vacuoles engulfing damaged mitochondria¹⁻³. Consistently, mitochondrial DNA copy was reduced in isolated retinal ECs from *Tfam*^{ECKO} mice, but not in *Cox10*- or *Trx2*-deficient retinal ECs. ATP contents were reduced in all mutant ECs (**Fig. 4m-n**). In combined with the in vitro data from siRNA silenced ECs (Fig. 1), our data suggest that the angiogenic retardation in the mutant mice was due to mitochondrial dysfunction rather resulted from massive cellular collapse in retinal ECs.

5- The arterialization of the vasculature should be validated by using intrinsic endothelial (arterial) cell markers such as Sox17 or Cx40.

-We performed immunostaining with an intrinsic arterial marker Sox17, which confirmed the increased arterialization in the mutant mice (Supplementary Fig. 5c-d).

6- scRNAseq is an elegantly approach which aims to provide some mechanistic understanding. This analysis would benefit by providing a better lay out of what it is different and what it is similar between the three models used.

- We have revised our presentation as follows:

1) Cluster analyses showed that most cells were EC populations (Clusters 1 to 9) and gene profiling revealed characteristics of each EC cluster common in three mutant mice (**Fig. 5a**). We then presented the cell numbers in each EC cluster of WT and three mutant mice separately (**Fig. 5b**). We also presented heatmaps in gene expression fold changes in each EC cluster of three mutant mice compared to WT mice (**Fig. 5c-h**; Supplemental Fig. 6b-d).

7- It would be interesting to validate whether EndMT occurs in vivo.

- First, we showed that the SMA⁺ microvessels were increased in the mutant retinas (**Fig. 4c**). Endothelial gain of SMA expression, a process of endothelial-mesenchymal transition (EndMT), has been implicated in many pathological diseases⁵. We then performed super resolution stimulated emission depletion (STED) microscopy. Our analyses showed that the SMA⁺ cells in the microvessels of *Tfam*^{ECKO} were mostly NG2⁺ pericytes and some CD31⁺ ECs (**Fig. 4d-f**), suggesting EndMT occurred in the mutant retinas.

8- The TGFb pathway has been extensively involved in the appearance of AVM. In this study, author identify a new family member (ALK5) in the involvement of mitochondrial-related AVM phenotype. However, the phenotype here relates to ALK5 gain of signalling instead of loss of TGFb function/signalling as it occurs in HHT. Is Alk5 a direct target of mitochondrial function or a consequence of the phenotype? Are the AVM and microaneurysms phenotypes reproduced by using an Alk5 GOF approach?

- We have provided evidence that mitochondrial dysfunction induces ALK5 expression at mRNA and protein levels. Moreover, this induction appeared to be dependent on MAPKs and the ALK5-SMAD2 signaling, suggesting that ALK5-SMAD2 is a direct target of mitochondrial dysfunction.

We appreciate your suggestion to use *Alk5* GOF to test if *Alk5* GOF induce similar microaneurysm phenotype. We couldn't find a good line with inducible *Alk5* GOF. Because

ALK5 and SMAD2 were increased in the mutant ECs, we have taken an alternative approach by testing if genetic deficiency of *Alk5* or *Smad2* could rescue the phenotype in the mutant mice. Deficiency of *Alk5*, but not of the canonical *Smad* effectors, cause retinal sprouting defects and hemorrhagic vascular malformations⁴. Therefore, we focused on genetic rescues by the *Smad* deficiency⁶. Our data showed that *Smad2* deficiency completely diminished the sprouting defects and microaneurysm formation in *Tfam*-KO mice (**Fig. 8**).

9- The manuscript is too lengthy (word wise) providing too many unnecessary details. Some of the parts could be simply improved by a making an effort with the wording.

- We have shortened the introduction and discussion to meet the 5,000-word limit policy.

10- Supp Fig 2b requires some revision, i.e. the CD31 and COX10/CD31 and TRX2/CD31 have to swapped positions, the TFAM stained images has some stars and numbers on top.

- We have corrected the labeling.

11- Figures 2g (lower panel on ERG/Edu) do not have enough resolution.

- We have provided better high-power images for ERG/Edu.

Response to Reviewer #2:

General comment: This is an important study that contributes significantly to our understanding of mitochondrial involvement in retinal angiogenesis. As far as I am aware, it is unique (note my lack of expertise in retinal vascular biology). The conclusions appear justified by the data presented. I detect no ethical compromises. The statistical analyses are straightforward.

-We appreciate your positive comments and your instructive suggestions.

Major comments:

1. This is a large study, part of which is not appropriate for a member of the Nature family journals directed at a general scientific audience. The section on postnatal developmental retinal vascular biology (mainly anatomy) is apparently well done, but very specialized. This material needs to be reviewed by someone with expertise in retinal vascular biology (certainly not me!) and published separately. I'll leave a decision about this point to the Editors.

- As Editors' suggested, we keep the retinal data.

2. The authors misused the "Supplemental Figures" section as an overflow presentation of many critical findings. I received a separate file of Supplemental figures, so it was relatively easy to read the manuscript and go back and forth from "regular" Figures to Supplemental Figures. I cannot envision a reader trying to navigate the manuscript easily and go back and forth as I did. The authors need to decide what is critical visual information, needing to be in the "regular" Figures, and what is truly "Supplemental" and not needed to prove the main points in the paper.

- There are limited 10 items in the main figures, therefore we have to put some data into Supplementary information.

3. I thought the material on mitochondrial dysfunction was well presented. While one can argue endlessly about the specificity (or lack thereof) of siRNA approaches (or even gene removal), these are established procedures.

- We have control siRNAs and controls mice.

4. I also liked the section on single-cell RNAseq, in which EC's from the tamoxifen-treated mice were enriched by flow cytometry, subjected to RNAseq and data reduced by a UMAP algorithm and clustered.

- Thank you very much.

5. The Discussion section is too long, in keeping with the paper being too large and unwieldy. It should be half as long at most, which will easily result if the paper is broken up.

- We have shortened the discussion.

6. Why are there no electroretinograms? A physiological correlate of the mitochondrial dysfunction (and vascular maldevelopment) would be contributory. It would seem to be helpful if ERG's were carried out in the absence and presence of treatment with the TGF beta R inhibitor.

- We appreciate your suggestion. We have tried the electroretinograms in our mice. However, it was quite challenging to perform the ERG assays in P6-P15 pups.

Minor comments:

1. There are several minor grammatical errors and mis-spellings. These are easily corrected.

- We have corrected these mis-spellings.

2. In at least two places the manuscript refers to "Leber's 'military' aneurysms", when the authors meant to say "Leber's miliary aneurysms". "DISCUSSION" is also mis-spelled.

- We sincerely apologize for the typos. We have corrected these mistakes.

Response to Reviewer #3

General comment: In my view, the current study is very-well designed and conducted.
- We appreciate your positive comments and your instructive suggestions.

Major concerns:

It is not clear how mitochondrial dysfunction involves or interferes with TGF β -ALK5-SMAD2 signaling?

- It appears that respiration defects represent a common signaling hub to activate downstream MAPK-SMAD2 signaling. Our data shows how mitochondria could act as a signaling organelles. Specifically, we show that mitochondrial dysfunction represents a common signaling hub to activate downstream MAPK-SMAD2 signaling, leading to similar phenotypes in the three mutant mice

Does TGF β R inhibitor also rescue ECs dysfunction in vitro (Figure 1)?

- We tested TGF β R (ALK5) inhibitors in a 3D sprouting assay. Silencing of *Tfam* significantly attenuated sprouting as indicated by the reduced number of sprouts and mean sprout length. Inhibition ALK5 signaling by LY364947 partially rescued *Tfam*-siRNA impaired sprouting as indicated by the increased sprout numbers and sprout lengths (Supplementary Fig. 8a-c).

One important issue is the central conclusion which represents TGF β R signaling as the major molecular mechanism causing vascular malfunction upon mitochondrial respiratory inactivation. This is mainly based on the use of a TGF β R inhibitor. Does an increased TGF β -ALK5-SMAD2 signaling under other experimental conditions cause the same phenotype?

- We appreciate your suggestion to verify the role of TGF β R signaling by other approaches beside the inhibitors. One way is use Alk5 GOF to test if Alk5 GOF induce similar microaneurysm phenotype. We couldn't find a good line with inducible Alk5 or Smad2 in retina. We couldn't find a good line with inducible Alk5 GOF. Because ALK5 and SMAD2 were increased in the mutant ECs, we have taken an alternative approach by testing if genetic deficiency of *Alk5* or *Smad2* could rescue the phenotype in the mutant mice. Deficiency of *Alk5*, but not of the canonical *Smad* effectors, cause retinal sprouting defects and hemorrhagic vascular malformations⁴. Therefore, we focused on genetic rescues by the *Smad* deficiency⁶. Our data showed that *Smad2* deficiency completely diminished the sprouting defects and microaneurysm formation in *Tfam*-KO mice (Fig. 8).

Minor points:

In the Introduction the description of the previous works dealing with the role of mitochondrial respiration in ECs is somehow inadequate. Authors stated:

However, recent data suggest that mitochondrial activity is critical for the biosynthetic and bioenergetic demands that are required for angiogenesis. Specifically, mitochondrial respiration, by providing the building blocks and metabolites necessary for EC proliferation rather than producing ATP, regulates angiogenesis at various steps 8,9. Schiffmann et al showed that mitochondrial respiration is important for EC growth and neoangiogenesis in embryonic development, wound healing and tumour growth. They use *cox10* KO mice for their study (similar to the current work).

- We have revised our Introduction (page 4).

Schiffmann et al showed that mitochondrial respiration is important for EC growth and neoangiogenesis in embryonic development, wound healing and tumour growth in EC-specific *Cox10* deficiency mice⁷.

Its regulatory role was concluded via the effects observed following the inhibition of the mitochondrial glucose metabolizing enzyme pyruvate dehydrogenase (PDHA1), the genetic deletion of fatty acid metabolizing enzyme mitochondrial carnitine palmitoyltransferase 1 (Cpt1), and glutamine limitation in 3D sprouting assays^{7,8,10-12}. More recently, a report showed that the loss of a subunit in complex III (QPC) caused a reduction in EC respiration that impaired cell proliferation and angiogenesis in mice¹³⁻¹⁵.

The cited works Chen et al ¹³ and Diaz et al ¹⁴ showed that Complex III alteration causes ROS production. These studies did not investigate any aspects of mitochondrial respiration in ECs. Instead, Schiffman et al ⁹ showed that EC-specific COX10 deficiency causes vascular phenotype and must be cited here rather than in the previous position. Coutelle et al (EMBO MO Med 2014, PMID: 24648500) also showed that mice harbouring defect in the proofreading function of the catalytic subunit of mitochondrial DNA (mtDNA) polymerase (POLGA) have an altered neoangiogenesis which could also be cited here as one of the initial studies using genetic mouse model to study the role of mitochondrial respiration in vascularization.

- We have revised our Introduction by correctly citing the previous work (page 4).

Coutelle et al showed that mice harbouring defect in the proofreading function of the catalytic subunit of mitochondrial DNA (mtDNA) polymerase (POLGA) had an altered angiogenesis⁸. The genetic deletion of fatty acid metabolizing enzyme mitochondrial carnitine palmitoyltransferase 1 (Cpt1) attenuated the formation and functioning of tip cells⁹. Similarly, loss of a subunit in complex III (QPC) caused a reduction in EC respiration that impaired cell proliferation and angiogenesis in mice¹⁰.

The phenotype of Tfam-, Cox10-, and Trx2-deficient mice have been characterized in various tissues and cell types; however, *little is known regarding their impacts on physiological angiogenesis*. Schiffmann et al used two independent Cre lines and showed that Cox10- in ECs is important for EC growth and neoangiogenesis in embryonic development, wound healing and tumour models.

- We have correctly cited the Schiffmann's work.

Treatment with LY in TfamEcko mice reduced phosphor-SMAD2 levels in ECs derived from these mice (Fig 7b). Increased phosphor-SMAD2 levels were detected in siTfam ECs (Figure 6d,e, g) but appeared to be rather unchanged in siCox10 ECs (Figure 6g, now is 6e) and were not further investigated. Then, how do the authors explain, that TGFbR inhibition also rescues the phenotype in Cox10Ecko mice?

- By quantifying the protein band of the initial blot, the phosphor-SMAD2 in siCox10 is ~2-fold higher than siCtrl ECs. We presented a better representative blot (see Fig.6d).

Response to Reviewer #4:

General comments: The authors demonstrated heterogeneous EC clusters by UMAP plots, selected GO terms significantly up/down-regulated in mutant mouse models by dotplots and expression of selected signature genes (Actin, ECM, Angiogenic, etc.) by heatmaps, trying to reveal the underlying mechanisms of AVM phenotype in three mutant mouse models. However, description of some results is confusing and needs to be stated more clearly.
- We thank you for your instructive comments and we have addressed your concerns as follows.

Major concerns:

1) For the scRNA-seq dataset, submission to a community-endorsed, public repository is mandatory. Accession numbers must be provided in the paper. (According to the Editorial policies of Nature portfolio: <https://www.nature.com/nature-portfolio/editorial-policies/reporting-standards#availability-of-data>). Moreover, without more detailed information (eg. gene expression profiling and metadata for single cells), I cannot make an objective and accurate judgment on some results shown here.

- As indicated by the editor, the concern has been addressed upon deposition of data online. Specifically, we have deposited our scRNA-seq data into GEO database with accession number (GSE172230). Count matrix and supplemental data is also provided as supplemental data from GEO. Currently, the data is not open to public, but is accessible by token, which is provided via editor.

2) GO terms shown in Fig. 5c are specifically selected or top most enriched by p values? Please specify it. If not top most enriched, consider to show them in the supplementary data.

- First of all, due to some redundancy of the original Fig.5a-c with Fig.5d-I, we deliberately deleted the original Fig.5a-c, but provided more detailed analyses on the EC clusters, including UMAP figures, cell numbers, and heatmaps of signature genes for EC clusters (Fig.5).

Fig.5 shows “representative” GO terms. GO analysis is good to annotate function, pathway and cellular events related to up- and down-regulated genes, but not perfect. Some significant GO terms are not directly related to endothelial functions. Therefore, we showed significant GO terms that are consistent with reduced microvessel or potentially important in retinal angiogenesis rather than simply showing top GO terms. Instead, we presented the whole-set of GO analyses in Supplemental Table 2.

3) Why did the authors switch from Seurat to destiny to further cluster ECs into nine subtypes? Are there any special considerations? How the batch effect (if it exists) was mitigated using destiny? As far as I know, diffusion map is usually used for Diffusion Pseudo Time analysis, and it is not common to be used for clustering. Please write more details of the clustering procedure in Methods. Besides, the authors stated nine EC subtypes were identified, but there is no further data to be shown to support their heterogeneity and characteristics of each subtype. It could be more informative to exhibit their DEGs (e.g. top 5 or 10) and/or GO terms in the Supplementary Figures or Tables.

- We apologize for the errors. We initially performed destiny from Seurat-normalized gene expression matrix, but the result was unclear and difficult to be interpreted. Therefore, we removed the destiny-based analysis from figures but we forgot to remove it from the method section. Instead, we used SCTransform normalization for EC subgrouping analysis. We revise the method description accordingly.

We presented the DEGs (top 20) for each cluster in Supplemental Table 1.

Method: EC subgrouping analysis was performed with SCTransform¹¹. Normalization and variance stabilization of single-cell RNA-seq data using regularized negative binomial regression} and Seurat¹². Highly Parallel Genome-wide Expression Profiling of Individual Cells Using

Nanoliter Droplets}. Briefly, SCtransform was implemented using percentage of mitochondria reads and total RNA count for linear regression.

4) Heatmaps displayed 1) average expression or fold change? 2) value of all ECs in one mouse model or only ECs included in the corresponding subtype (eg. Actin genes in Fig. 5e for Cluster 1)? Please describe these more clearly. If all ECs, whether distribution differences of different mouse models will bias the results? For example, Ribosome genes were highly expressed in three mutant models compared to WT, given that less ECs were in cluster 7 in WT. If not, please state more clearly in the text. Similarly, there are same concerns for Fig. 6a-c and Supplementary Fig. 6b-d. In addition, it seems that Fig. 5e-i were cited incorrectly in the main text (Page 13).
- We have clarified that 1) Fold changes of gene expression in the mutant ECs vs WT were presented in Fig. 5 and Fig.6; 2) Given the EC distribution is different in each cluster, heatmaps were generated by including all ECs in each model. We have now correctly cited for Fig.5 in the main text.

5) I noticed that nine EC subtypes might be differentially distributed in four mouse models (Fig. 5d). This would be clearer to show a bar plot of distributions of EC subtypes in each mouse model. Whether the differences in distributions of EC subtypes are also associated with the phenotype of mutant mouse?

- We appreciate your valuable comments. We have now provided a bar plot showing the distribution of EC subtypes in each mouse model (Fig. 5b).

6) The authors only showed Fold Changes in the heatmaps (Fig. 5e-i, Fig. 6a-c and Supplementary Fig. 6b-d). Statistical significance should also be shown, and asterisk symbols indicated in each tile is recommended.

- Statistical significances are presented in Fig.5 and Fig.6 with asterisk symbols as you suggested.

References cited in this response:

1. Huang, Q., *et al.* Thioredoxin-2 inhibits mitochondrial reactive oxygen species generation and apoptosis stress kinase-1 activity to maintain cardiac function. *Circulation* **131**, 1082-1097 (2015).
2. He, F., *et al.* Mitophagy-mediated adipose inflammation contributes to type 2 diabetes with hepatic insulin resistance. *J Exp Med* **218**, e20201416 (2021).
3. Huang, Y., *et al.* Brown adipose TRX2 deficiency activates mtDNA-NLRP3 to impair thermogenesis and protect against diet-induced insulin resistance. *J Clin Invest* (2022).
4. Zarkada, G., *et al.* Specialized endothelial tip cells guide neuroretina vascularization and blood-retina-barrier formation. *Developmental cell* **56**, 2237-2251 e2236 (2021).
5. Kalluri, R. & Weinberg, R.A. The basics of epithelial-mesenchymal transition. *J Clin Invest* **119**, 1420-1428 (2009).
6. Ju, W., *et al.* Deletion of Smad2 in mouse liver reveals novel functions in hepatocyte growth and differentiation. *Mol Cell Biol* **26**, 654-667 (2006).
7. Schiffmann, L.M., *et al.* Mitochondrial respiration controls neoangiogenesis during wound healing and tumour growth. *Nat Commun* **11**, 3653 (2020).
8. Coutelle, O., *et al.* Embelin inhibits endothelial mitochondrial respiration and impairs neoangiogenesis during tumor growth and wound healing. *EMBO Mol Med* **6**, 624-639 (2014).
9. Yetkin-Arik, B., *et al.* The role of glycolysis and mitochondrial respiration in the formation and functioning of endothelial tip cells during angiogenesis. *Sci Rep* **9**, 12608 (2019).
10. Diebold, L.P., *et al.* Mitochondrial complex III is necessary for endothelial cell proliferation during angiogenesis. *Nat Metab* **1**, 158-171 (2019).
11. Hafemeister, C. & Satija, R. Normalization and variance stabilization of single-cell RNA-seq data using regularized negative binomial regression. *Genome Biol* **20**, 296 (2019).
12. Macosko, E.Z., *et al.* Highly Parallel Genome-wide Expression Profiling of Individual Cells Using Nanoliter Droplets. *Cell* **161**, 1202-1214 (2015).

Reviewers' Comments:

Reviewer #1:

Remarks to the Author:

With all the additions and clarifications, the manuscript has improved substantially. The genetic rescue makes the story more compelling and complete. I only have one minor point: I am not convinced that ECs are α SMA positive as the authors claim. (Fig. 4d). Perhaps the authors can check if the EndoMT cluster in the scRNAseq expresses α SMA. This would reinforce this notion.

Reviewer #2:

Remarks to the Author:

This is a revision of a manuscript previously submitted to Nature Communications in which the authors diminish in an endothelial cells (EC)-specific manner the activities (usually but not always by gene deletion) of several important mitochondrial proteins coded in the nuclear genome of mice and study the effects of these changes on retinal vascular development. The original manuscript has been altered by additional experimental detail and shortening of the Discussion. These are meaningful changes to the original manuscript, that was important for its implications as to energy needs of developing retinal vasculature under normal circumstances, and the implication that deficient energy transformations may underlie one or more retinal vascular conditions. The authors also demonstrate the possibility that the mitochondrial protein alterations may disrupt retinal EC's and vascular development by altering ALK5-SMAD2 signaling.

Although the manuscript is improved, my two original concerns remain. The first concern is that there are too many Supplemental Figures, particularly when several of them display important primary data. I feel that with careful attention to contents, at least half of the Supplemental figures could be combined with "real" Figures. My second (and perhaps newer) concern relates to the uses of the non-specific terms "mitochondrial activity" and "mitochondrial dysfunction". The authors are altering specific mitochondrial proteins (Tfam, COX10, Trx2) and show (see Fig 4 k-n) that all of these manipulations alter ATP levels, mitochondrial anatomy and mitochondrial number. I did not find any analysis of mitochondrial size, as both fission and fusion of mitochondria are energy-requiring processes that can alter apparent "number". I encourage the authors to be more specific.

A minor point is that I feel the use of "dive", "diving", etc is inappropriate as it implies intentionality of EC's. Rather, "migration to deeper levels", or equivalent terms would seem better.

Reviewer #3:

Remarks to the Author:

Authors answered almost all my question. I was however hoping to have their views on the molecular link between mitochondria and TGF β -ALK5-SMAD2 signalling. Mentioning that mitochondria just act as signalling hubs does not answer this question. I believe that this question can not be properly handled in the frame of the current study and therefore would like to thank authors for providing such extensive study to show how important mitochondria are in EC function.

Reviewer #4:

Remarks to the Author:

Using scRNA-seq, the authors defined heterogeneous EC subgroups, and compared the differences between WT and mutant mouse models among subgroups by heatmap. However, description of some results is confusing and needs to be stated more clearly. Major comments are listed as follows:

1. Enrichment of mitochondrial-related genes were shown in the heatmap of supplementary fig. 6b and in the top20 DEGs of cluster3 supplementary table 1. However, in the part of data quality

control, the author only excluded cells with less than 500 genes for downstream analysis. Is this difference caused by low cell activity? It is necessary to evaluate mitochondrial gene expression and further filter cells with poor activity. The relevant analysis should be described in detail in the supplementary data.

2. In fig5a, the author showed the defined 0-10 clusters with umap and named PC, Actin, Mito, TGF β What is the evidence for naming these clusters? Although top20 DEGs were shown in the supplementary table 1, the author does not provide more description and analysis to describe the heterogeneity and characteristics of each EC subtype. More data should be added for further illustration (such as characteristic gene expression or GO terms). In fact, the whole-set of GO analysis in Supplementary Table 2 is not friendly for making the readers understand.

3. The major problem in the analysis is that, as indicated in fig5 c-h, fig6a, and supplementary fig 6b, "Heatmap of signature genes representing EC clusters. Given the EC distribution is different in each cluster, heatmaps were generated by including all ECs in each model". However, in the manuscript (Page10), these were described as each cluster but not all ECs. Extending the characteristics of a cluster to the whole ECs does not seem reasonable. Instead of defining EC subgroups, it would make more sense to describe the molecular alterations of mutant mouse ECs through unbiased comparison of DEGs or GO terms between different mutant mouse models and WT. In addition, it is essential to visualize the expression of significant genes through violin/dot plots, heatmaps, etc.

POINT-TO-POINT RESPONSE TO REVIEWERS' COMMENTS

RESPONSE TO REVIEWER 1

With all the additions and clarifications, the manuscript has improved substantially. The genetic recuse makes the story more compelling and complete. I only have one minor points: I am not convinced that ECs are aSMA positive as the authors claim. (Fig. 4d). Perhaps the authors can check if the EndoMT cluster in the scRNAseq expresses aSMA. This would reinforce this notion.

- Yes, you are right about the aSMA expression in EC. scRNA-seq analyses indicated that aSMA (*Acta2* gene) was enriched in pericyte cluster, but not in EC clusters. We have revised the text on EndMT accordingly.

Page 9: Super resolution stimulated emission depletion (STED) microscopy analyses showed that the SMA⁺ cells in the microvessels of *Tfam*^{ECKO} were mostly NG2⁺ pericytes with few CD31⁺ ECs (**Fig. 5d-f**), suggesting a minimum of EndMT occurred in the mutant retinas.

RESPONSE TO REVIEWER 2

This is a revision of a manuscript previously submitted to Nature Communications in which the authors diminish in an endothelial cells (EC)-specific manner the activities (usually but not always by gene deletion) of several important mitochondrial proteins coded in the nuclear genome of mice and study the effects of these changes on retinal vascular development. The original manuscript has been altered by additional experimental detail and shortening of the Discussion. These are meaningful changes to the original manuscript, that was important for its implications as to energy needs of developing retinal vasculature under normal circumstances, and the implication that deficient energy transformations may underlie one or more retinal vascular conditions. The authors also demonstrate the possibility that the mitochondrial protein alterations may disrupt retinal EC's and vascular development by altering ALK5-SMAD2 signaling.

Although the manuscript is improved, my two original concerns remain. The first concern is that there are too many Supplemental Figures, particularly when several of them display important primary data. I feel that with careful attention to contents, at least half of the Supplemental figures could be combined with "real" Figures. My second (and perhaps newer) concern relates to the uses of the non-specific terms "mitochondrial activity" and "mitochondrial dysfunction". The authors are altering specific mitochondrial proteins (Tfam, COX10, Trx2) and show (see Fig 4 k-n) that all of these manipulations alter ATP levels, mitochondrial anatomy and mitochondrial number. I did not find any analysis of mitochondrial size, as both fission and fusion of mitochondria are energy-requiring processes that can alter apparent "number". I encourage the authors to be more specific.

- Thank you very much for your suggestions. We have moved some of the supplementary data into the main figures. Specifically, we have incorporated S1 to Fig.1; S7 to Fig.7. Total figures have 10. However, due to limited numbers and space for the main figures, we can only incorporate a few.

Thank you for your suggestions. We have used "mitochondrial dysfunction" as a general term to indicate mitochondria in cells exhibit structural or/and activity changes, including ATP production, ROS generation, mitochondrial anatomy and mitochondrial numbers.

In addition, we have examined mitochondrial fission and fusion. However, we found both fission and fusion pathways were altered in the mutant ECs. Similarly, both larger and smaller mitochondria were detected in the mutant ECs. These data suggest that mitochondrial dynamics and biogenesis might be altered. We need to perform more analyses in our future studies to draw clear conclusions. Thank you for your understanding.

A minor point is that I feel the use of "dive", "diving", etc is inappropriate as it implies intentionality of EC's. Rather, "migration to deeper levels", or equivalent terms would seem better.

- We have revised "dive" to "migration to deeper layers" as you suggested.

RESPONSE TO REVIEWER 3

Authors answered almost all my question. I was however hoping to have their views on the molecular link between mitochondria and TGF β -ALK5-SMAD2 signalling. Mentioning that mitochondria just act as signalling hubs does not answer this question. I believe that this question cannot be properly handled in the frame of the current study and therefore would like to thank authors for providing such extensive study to show how important mitochondria are in EC function.

- Thank you very much for your understanding and your positive comment that “would like to thank authors for providing such extensive study to show how important mitochondria are in EC function”.

RESPONSE TO REVIEWER 4

Using scRNA-seq, the authors defined heterogeneous EC subgroups, and compared the differences between WT and mutant mouse models among subgroups by heatmap. However, description of some results is confusing and needs to be stated more clearly. Major comments are listed as follows:

1. Enrichment of mitochondrial-related genes were shown in the heatmap of supplementary fig. 6b and in the top20 DEGs of cluster3 supplementary table 1. However, in the part of data quality control, the author only excluded cells with less than 500 genes for downstream analysis. Is this difference caused by low cell activity? It is necessary to evaluate mitochondrial gene expression and further filter cells with poor activity. The relevant analysis should be described in detail in the supplementary data.

- Thank you very much for your suggestion. Based on your suggestion, the scRNAseq data was reanalysed and, as such, the corresponding methods and results sections were rewritten to provide more details. This time, we have also filtered cells with poor activity, and the relevant analysis is now described in detail in the Method section (page 24). Upon this filtration, the mitochondrial cluster is no longer detected as a separate cluster. Please see revised Figure 6 and the corresponding text (page 9-10).

2. In fig5a, the author showed the defined 0-10 clusters with umap and named PC, Actin, Mito, TGFβ.... What is the evidence for naming these clusters? Although top20 DEGs were shown in the supplementary table 1, the author does not provide more description and analysis to describe the heterogeneity and characteristics of each EC subtype. More data should be added for further illustration (such as characteristic gene expression or GO terms). In fact, the whole-set of GO analysis in Supplementary Table 2 is not friendly for making the readers understand.

- We have now provided more detailed description on the Clusters. First, a total 10 clusters were identified. Based on cell type-specific marker genes, two segregated smaller clusters were identified as peryctes (PC; Cluster 8) using markers *Pdgfrb* and *Cspg4* and photoreceptor cells (Cluster 9) using *Gngt1* and *Sag* markers. These two clusters were excluded for further analyses. Secondly, we annotated the rest of EC clusters (cluster 0-7) based on visual inspection (Supplementary table 1) and GO analysis (Supplementary Table 2) of the positive differentially expressed genes for each cluster compared to all other cells. Of note, Cluster 3 for BBB was primarily WT ECs whereas Cluster 4 for ECM were largely the mutant ECs without WT ECs (**Fig.6a-b**).

As mentioned before, we have now rewritten the scRNAseq Methods (please, see page 24) and Results (please, see page 9-10) sections to indicate these changes.

3. The major problem in the analysis is that, as indicated in fig5 c-h, fig6a, and supplementary fig 6b, "Heatmap of signature genes representing EC clusters. Given the EC distribution is different in each cluster, heatmaps were generated by including all ECs in each model". However, in the manuscript (Page10), these were described as each cluster but not all ECs. Extending the characteristics of a cluster to the whole ECs does not seem reasonable. Instead of defining EC subgroups, it would make more sense to describe the molecular alterations of mutant mouse ECs through unbiased comparison of DEGs or GO terms between different mutant mouse models and WT. In addition, it is essential to visualize the expression of significant genes through violin/dot plots, heatmaps, etc.

- Thank you for your suggestions. We have performed unbiased comparisons of differential expressed genes (DEGs) between the different mutant mouse models and WT. Specifically, DEGs in the total EC populations (Cluster 0-7 combined together upon removing pericyte and neuron clusters) were compared between each mutant with the WT (see GEO files [Tfam, Cox10 or Trx2]_vs_WT_DE_SCT_selectedClusters.xlsx). We identified 138 commonly regulated genes (Supplementary Table S3). Interestingly, these genes were similarly upregulated or downregulated in the mutant ECs compared to WT (**Fig. 6c**). Among these 138 genes, transporter genes were drastically downregulated in the mutant ECs (**Fig. 6d**), consistent with immature vasculature in the mutant retinas. On the other hand, the ECM-associated genes were either highly upregulated or significantly downregulated in the mutant ECs (**Fig. 6e**). Moreover, we found that the TGF β regulatory pathway, a major regulator of ECM-associated genes, was commonly altered in the three mutant ECs (**Fig. 6f**). We also have extended the results and discussion section to incorporate these findings.

Reviewers' Comments:

Reviewer #4:

Remarks to the Author:

The revised version of the manuscript is better understandable. Some minor suggestions are listed below:

1. In fig. 6, the authors annotated eight EC clusters (cluster 0-7) based on visual inspection (Supplementary table 1) and GO analysis (Supplementary Table 2). It is recommended that the author complement the expression of related genes (with dotplot or volinplot) and GO terms in the figures, instead of just showing the names of EC subclusters, so that readers can understand the heterogeneity of each EC subgroup.
2. In the results (page 10-11), the author stated that "We further determined if the TGF β signaling was altered in the mutant retinas. Immunostaining indicated that phosphor-SMAD2 was highly upregulated in the retinal ECs in the mutant retinas". Is there any difference in Smad2 expression between mutant retinas and the WT in scRNA-seq dataset? It doesn't seem to be presented in fig. 6 or the DEGs.
3. In fig. 6a and 6b, the same cluster uses different colors, which may cause misunderstanding. It will be better if the colors are consistent.
4. In Supplementary Tables 2 and 3, it is sug

POINT-TO-POINT RESPONSE TO REVIEWERS' COMMENTS

RESPONSE TO REVIEWER 4

The revised version of the manuscript is better understandable. Some minor suggestions are listed below:

- Thank you very much for your positive response. We have addressed your concerns as below.

1. In fig. 6, the authors annotated eight EC clusters (cluster 0-7) based on visual inspection (Supplementary table 1) and GO analysis (Supplementary Table 2). It is recommended that the author complement the expression of related genes (with dotplot or volinplot) and GO terms in the figures, instead of just showing the names of EC subclusters, so that readers can understand the heterogeneity of each EC subgroup.

- We have provided the expression of related genes with dotplot (Fig. 6b) and GO terms (Fig. 6c) in the figures. The text and the figure legend have been revised accordingly (pages 10-11).

2. In the results (page 10-11), the author stated that "We further determined if the TGF β signaling was altered in the mutant retinas. Immunostaining indicated that phosphor-SMAD2 was highly upregulated in the retinal ECs in the mutant retinas". Is there any difference in Smad2 expression between mutant retinas and the WT in scRNA-seq dataset? It doesn't seem to be presented in fig. 6 or the DEGs.

- Based on the scRNA-seq analyses, Smad2 mRNAs was significantly increased in *Tfam* and *Trx2*-deficient ECs but not in *Cox10*-deficient ECs, therefore it was not presented in the Fig. 6 or the common DEGs. The highly upregulated phosphor-SMAD2 in the retinal ECs of the mutant retinas represented increased activation of SMAD2.

3. In fig. 6a and 6b (now 6d), the same cluster uses different colors, which may cause misunderstanding. It will be better if the colors are consistent.

- We have revised the Fig. 6d so the colors are consistent with Fig. 6a.

4. In Supplementary Tables 2 and 3, it is suggested to provide the complete information, including P-value, log 2-fold changes, etc.

- We have now included $-\log_{10}(\text{FDR})$ in Supplementary Table 2, and log 2-fold changes and P-values in Supplementary Table 3.